# Molecular Epidemiology and Evolutionary Dynamics of Human Influenza Type-A Viruses in Africa: A Systematic Review

**DOI:** 10.3390/microorganisms10050900

**Published:** 2022-04-25

**Authors:** Grace Nabakooza, Ronald Galiwango, Simon D. W. Frost, David P. Kateete, John M. Kitayimbwa

**Affiliations:** 1Department of Immunology and Molecular Biology, Makerere University, Old Mulago Hill Road, P.O. Box 7072, Kampala 256, Uganda; davidkateete@gmail.com; 2UVRI Centre of Excellence in Infection and Immunity Research and Training (MUII-Plus), Makerere University, Plot No: 51-59 Nakiwogo Road, P.O. Box 49, Entebbe 256, Uganda; galiwango.gronald.ronald@gmail.com (R.G.); kittsra@gmail.com (J.M.K.); 3Centre for Computational Biology, Uganda Christian University, Plot 67-173, Bishop Tucker Road, P.O. Box 4, Mukono 256, Uganda; 4African Center of Excellence in Bioinformatics and Data Intensive Sciences, Infectious Diseases Institute, Makerere University, Kampala 256, Uganda; 5Microsoft Research, Redmond, 14820 NE 36th Street, Washington, DC 98052, USA; frost.simon@microsoft.com; 6London School of Hygiene & Tropical Medicine (LSHTM), University of London, Keppel Street, Bloomsbury, London WC1E7HT, UK

**Keywords:** influenza type-A viruses, seasonal influenza, pandemic influenza, genomic diversity, antigenic drift, drug resistance, reassortment, phylogenetics, phylodynamics, phylogeography, sub-Saharan, Africa

## Abstract

Genomic characterization of circulating influenza type-A viruses (IAVs) directs the selection of appropriate vaccine formulations and early detection of potentially pandemic virus strains. However, longitudinal data on the genomic evolution and transmission of IAVs in Africa are scarce, limiting Africa’s benefits from potential influenza control strategies. We searched seven databases: African Journals Online, Embase, Global Health, Google Scholar, PubMed, Scopus, and Web of Science according to the PRISMA guidelines for studies that sequenced and/or genomically characterized Africa IAVs. Our review highlights the emergence and diversification of IAVs in Africa since 1993. Circulating strains continuously acquired new amino acid substitutions at the major antigenic and potential N-linked glycosylation sites in their hemagglutinin proteins, which dramatically affected vaccine protectiveness. Africa IAVs phylogenetically mixed with global strains forming strong temporal and geographical evolution structures. Phylogeographic analyses confirmed that viral migration into Africa from abroad, especially South Asia, Europe, and North America, and extensive local viral mixing sustained the genomic diversity, antigenic drift, and persistence of IAVs in Africa. However, the role of reassortment and zoonosis remains unknown. Interestingly, we observed substitutions and clades and persistent viral lineages unique to Africa. Therefore, Africa’s contribution to the global influenza ecology may be understated. Our results were geographically biased, with data from 63% (34/54) of African countries. Thus, there is a need to expand influenza surveillance across Africa and prioritize routine whole-genome sequencing and genomic analysis to detect new strains early for effective viral control.

## 1. Introduction

Influenza type-A viruses (IAVs) cause potentially deadly respiratory infections among humans globally [1]. The evolution of IAVs is punctuated by gene exchanges (reassortment) between avian, swine, and human IAVs responsible for the past global influenza pandemics: the 1918-H1N1, 1957-H2N2, 1968-H3N2, and the 2009-H1N1pdm09 that claimed millions of lives [1,2,3,4]. The effects of the pandemics are further worsened by seasonal influenza infections estimated to cause excess mortality at a rate of 4.0–8.8 per 100,000 individuals globally, with the highest rates (2.8–16.5 per 100,000 individuals) in sub-Saharan Africa [5].

The influenza type-A viral whole-genome consists of eight single-stranded RNA gene segments coding for two immune-stimulating surfaces (hemagglutinin-HA and neuraminidase-NA), one transmembrane, and five internal proteins [6]. There are 18 HA and 11 NA subtypes of IAVs, but frequently H1-H3 and N1-N2 viruses have been sampled among humans [7].

Genomic characterization of circulating IAVs to understand their evolution and how novel viruses emerge and spread among populations is important for effective viral control and prevention. However, the rapid and continuous genomic changes through mutations (genes) or substitutions (proteins) [8], reassortment, and rarely recombination [9] make the predictions very difficult. Amino acid substitutions at the major antigenic sites of the IAV H1 (Sa, Sb, Ca_1_, Ca_2_, and Cb) and H3 proteins (A, B, C, D, and E) [10,11,12], and the NA proteins [13,14] cause the IAVs to drift away from vaccine-mediated immunity. IAVs also escape host immunity by gaining new N-linked glycosylation sites, Asn-X-Ser/Thr, where X is any amino acid except proline, in their HA and NA proteins [15,16]. When a sugar molecule attaches to the amide nitrogen on the Asn, the HA antigenic and NA catalytic sites become masked from binding to host antibodies and protected, respectively [15]. Substitutions in the internal genes of IAVs also result in new viruses with altered pathogenicity, virulence, transmissibility, and reduced sensitivity to antiviral drugs and vaccines [17,18,19].

Modern sequencing technologies, bioinformatic and phylogenetic-based analyses, and epidemiological and evolution models have successfully been used on viral genome data from well-sampled regions in Asia, Europe, and the United States (U.S.) to infer viral genomic and antigenic evolution, diversification, and transmission patterns [20,21,22,23,24,25,26,27]. Contrary to common belief, influenza surveillance in Africa has improved substantially since the mid-2000s, showing the cocirculation of multiple viral subtypes all year round with distinct peaks in Northern and Southern Africa [28,29,30]. However, data on the genomic characterization of IAVs in Africa remain scanty, limiting our understanding of Africa’s IAVs evolution, contribution to the global influenza ecology, and benefit from viral control strategies. We sought to systematically merge data on the evolution, diversification, drug resistance, zoonosis, phylogenetic clustering and genetic clades, phylodynamics, and phylogeographic patterns of H1N1, H1N1pdm09, and H3N2 virus strains previously sampled in Africa.

## 2. Materials and Methods

### 2.1. Definitions and Outcomes

We defined a subtype as a group of viruses with a specific combination of HA and NA genes, for example, H1N1 and H3N2 [7]. A viral lineage is a group of genetically related viruses of the same subtype sharing a common ancestor. A strain is a virus exhibiting minor mutations different from other viruses of the same lineage or subtype.

A phylogeny is a tree-like structure reconstructed using viral gene sequences showing evolutionary relationships between the sampled viruses from a common ancestor. The tree has internal nodes, branches, and tips representing ancestors, genetic distances from the nearest ancestors, and sampled viruses, respectively.

Phylogenetic clusters and clades are often interchanged [31]. Here, we defined a cluster as a subtree of viruses sharing a common ancestor and may share epidemiological linkage [32]. In contrast, a clade is a cluster of viruses with characteristic amino acid substitutions in their HA proteins subunits (HA1 or HA2) [27]. Phylodynamics is the study of evolutionary processes that shape the phylogenies, such as changes in viral effective population sizes [*Ne*(*t*)], evolutionary rate, basic reproductive number (*R*_0_), selection pressures, and reassortment [25,33]. Additional definitions are provided in Appendix A.

### 2.2. Search Strategy and Selection Criteria

We searched 7 databases: African Journals Online, Embase, Global Health, Google Scholar, PubMed, Scopus, and Web of Science on 30 July 2021 for published articles on sequencing and genomic characterization of IAVs sampled in Africa, according to the PRISMA guidelines [34]. The search keywords included (“influenza”) AND (“sequencing” OR “genomic analysis” OR “phylogenetic analysis”) and their respective medical subject heading (MeSH) terms, AND (each African country, region, and the Indian Ocean) NOT (non-human influenza or other pathogens causing influenza-like-illnesses), as provided in Appendix A. We did not filter by language or publication date.

All searched study articles were exported to and deduplicated using Zotero citation manager (https://www.zotero.org/, last accessed on 30 July 2021). Non-English articles were translated using google translate (https://translate.google.com, last accessed on 30 July 2021). Unique articles were subjected to a primary screening for eligibility based on the title, abstract, and method using a predefined eligibility criterion (Appendix A). We excluded studies that did not confirm influenza infection by culture or/and polymerase chain reaction (PCR), analyzed only non-human or other human influenza viruses (type B, C, and D), case reports, conference abstracts and presentations, health meetings, and sampled only non-African populations.

Eligible articles identified through the primary screening were subjected to a secondary search by screening their bibliographies for additional relevant articles. Authors were contacted for details on studies with sufficient information to decide on their inclusion. Eligible studies were agreed upon by G.N., R.G., J.M.K., and D.P.K.

### 2.3. Data Extraction and Synthesis

G.N. and R.G. extracted study data using a predefined data collection form. Data extracted for each study included: country, sampling period and setting, population demographics, selection criteria of samples for sequencing or pre-collected sequences, sample size, sequencing technology, number of viruses sequenced, genes sequenced or/and analyzed, analysis type, epidemiological variables linked to sequence data, and main findings.

Extracted data were stratified according to viral subtype and analysis type. We systematically report results from the earliest to the latest sampling year to highlight differences in viral evolution and transmission patterns between seasons.

### 2.4. Study Quality Assessment

Eligible studies were assessed for quality and risk-of-bias using the Newcastle-Ottawa Quality Assessment Scale-NOS (adapted for cross-sectional studies) [35] with additional items from the Strengthening the Reporting of Molecular Epidemiology for Infectious Diseases (STROME-ID) checklist [36]. Studies were evaluated based on the sampling, reporting missing data, outcome assessment, and result selection and reporting (Appendix A).

The review was reported according to the preferred reporting items for systematic review (PRISMA) guidelines, and its protocol was registered on PROSPERO (https://www.crd.york.ac.uk/PROSPERO/, last accessed on 23 March 2022) under ID CRD42020187011.

## 3. Results

### 3.1. Search Outcomes and Study Characteristics

The primary search identified 1337 study articles, 60 of which were eligible (Figure 1). The secondary search identified 11 additional eligible articles, bringing the total to 71 articles included in the final analysis.

A total of 70% (50/71) of the included studies were nested within surveillance programs, and 30% (21/71) were independent (Table 1). The included studies analyzed IAVs sampled from 34 (63%) of the 54 African countries (Appendix A), with most studies characterizing viruses from South Africa (*n* = 42), Kenya (*n* = 25), and Ghana (*n* = 24) (Figure 2). A total of 14, 41, and 42 studies characterized H1N1, H1N1pdm09, and H3N2 viruses sampled for 14 (1996–2009), 10 (2009–2018), and 26 (1993–2018) years, respectively.

A total of 87% (62/71) of the studies generated and analyzed new viral sequences, and 13% (9/71) analyzed online sequences. Of the studies that generated new sequences, 53% (33/62) and 37% (23/62) sequenced IAVs locally and abroad, especially in the United Kingdom (U.K.) and the United States (U.S.), respectively. Two studies sequenced IAVs both locally and abroad, and four studies did not report where sequencing was performed.

A total of 56% (35/62) of the studies used Sanger sequencing technology and were published between 1996 and 2020. Only 8% (5/62) of the studies used next-generation sequencing (NGS) and were published between 2011 and 2021 [75,76,81,97,98]. Two studies used both Sanger and NGS [26,73]. Another study did not clearly report whether Sanger or NGS was used [94]. Three studies used pyrosequencing or Sanger and pyrosequencing or Sanger and targeted HA analysis. Sixteen studies did not describe the technology used.

A total of 82% (58/71) of included studies sequenced and/or analyzed partial genomes, especially the HA and/or NA and/or MP. A total of 16% (11/71) studies sequenced and/or analyzed whole genomes [26,44,47,54,55,57,63,73,75,95,98]. One study sequenced whole genomes but analyzed only the HA genes [81]. Two studies analyzed only MP or NS genes. Viral phylogenetic patterns were the most characterized by 77% (55/71) of included studies followed by 68% (48/71) studies reported on virus-vaccine genomic variation. Viral drug sensitivity and phylodynamics were described in 39% (28/71) and 17% (12/71) studies, respectively. One study [100] had data on viral recombination.

### 3.2. Quality of Studies Analyzed

Our quality and risk-of-bias assessment scheme scored each study out of 23 points (Appendix A). Given the additional assessment terms, the analyzed studies had a mean quality score of 54.9% (30.4–73.9) (Appendix A), lower than the 90% (70–100) reported by Modesti [35]. The detailed assessment of the sampling bias for all included studies is provided in Appendix A.

### 3.3. Molecular Epidemiology of Seasonal H1N1 Viruses in Africa

#### 3.3.1. Genomic Diversity of H1N1 Viruses and Their Relatedness to Vaccines

Africa H1N1 viruses evolved every season. Virus strains sampled before 2007 exhibited minor genomic variation from their corresponding season vaccine strains. For example, five Morocco H1N1 strains sampled between 1996 and 1997 had a single substitution V57I in their HA1 proteins relative to the 1997–1998 season A/Bayern/7/1995(H1N1) vaccine strain and were recognized by antisera against the previous 1987–1996 northern hemisphere (NH) and 1988–1997 southern hemisphere (SH) season A/Singapore/6/1986(H1N1) vaccine strain [83]. Strains sampled later in Dakar (Senegal) during the 1997–1999 and 2000–2003 seasons evolved and were genetically and antigenically similar to the A/Beijing/262/1995(H1N1) (1998–2000 NH and 1999 SH season) and A/New Caledonia/20/1999(H1N1) (2000–2007 NSH season) vaccine strains, respectively [89]. New H1N1 strains similar to the 2008–2010 NH and 2009 SH season A/Brisbane/59/2007(H1N1) vaccine strain emerged in Dakar [89], Kenya [38], and South Africa [37] between 2007 and 2008. While Kenya H1N1 strains sampled in 2007 had three mutated antigenic sites Sb (D190N/G) and Ca_2_ (D225G), more substitutions at site Sb (N187S, G189N/A, K192R, A193T) and receptor-binding sites (N187S, G189N, A193T) were observed in 2008 relative to A/Brisbane/59/2007(H1N1) strain [38].

#### 3.3.2. Antiviral Drug Sensitivity among H1N1 Viruses

All H1N1 strains that circulated in Egypt, Kenya, and Senegal between 2004 and 2009 lacked the adamantane-resistant marker S31N in their M2 proteins [41,87,105]. The adamantane-susceptible strains sampled in Kenya between 2008 and 2009 had HA1 substitutions R192H, K140E, K145R, and N183S absent in global adamantane-resistant H1N1 strains (with E140K) [105].

Oseltamivir-resistant H1N1 strains with N1 substitution H275Y (H274Y, N2 numbering) emerged at different times in 2008 and circulated at high frequencies (70–100%) in several African countries. The first H1N1-H275Y variant was sampled in Cameroon in late February, Ivory Coast and Seychelles in April, South Africa in May, and Senegal in July 2008 [37,39,40,41,88,104,105]. The H1N1-H275Y variants had additional N1 substitutions unique to the country of sampling: South Africa (M23L, N73K) [37], Senegal (S141N, G185A) [41], and Cameroon, Ivory Coast, and Seychelles (D354G) [40,88]. Neuraminidase inhibition (NAI) assays confirmed that Africa H1N1-H275Y variants were resistant to oseltamivir and peramivir but susceptible to zanamivir [37,39,40,104].

#### 3.3.3. Phylogenetic Clustering Patterns and Circulating Clades among H1N1 Viruses

Phylogenies of Africa H1N1 virus strains showed the cocirculation of multiple viral lineages, clusters, and clades from 1996 through 2009. South Africa H1N1 strains sampled in 1996 were highly clustered (bootstrap > 98) with European-origin A/Bayern/7/1995(H1N1)-like strains [85]. An Asian-origin A/Wuhan/371/1995(H1N1)-like lineage emerged in 1997 and co-circulated with the A/Bayern/7/1995 (H1N1)-like strains in South Africa up to 1999 [85]. Similarly, Asian-origin A/Beijing/262/1995(H1N1)-like strains emerged in Dakar in 1997 and circulated through 1999 [89]. In 2001, novel Oceania-origin A/New Caledonia/20/1999(H1N1)-like strains belonging to clade 1 emerged in Egypt and were also sampled later in Senegal and Reunion (2003), Morocco (2004), South Africa (2005), Egypt (2005 and 2006), and Madagascar (2006) [89] (Figure 3). During the same season (2001), novel reassortant H1N2 viruses with HA1 substitutions V175I, A190T, and A215T emerged in Egypt and co-circulated with H1N1 viruses [89]. H1N2 strains were later sampled in South Africa (2002) and Senegal (2003) [89] but disappeared shortly after their emergence.

A South Africa strain, A/Johannesburg/141/2007(H1N1), sampled in 2007, shared a similar H1 gene as an American-origin A/St.Petersburg/10/2007-like(clade 2C) strain with HA1 substitutions T82K, Y94H, and K141E [89]. In the same year (2007), new Australian-origin A/Brisbane/59/2007(H1N1)-like strains (clade 2B, mostly oseltamivir-resistant) with HA1 substitutions (D35N, R159K, E274K) emerged in Egypt, Madagascar, Mauritius, and Senegal [41,89]. Clade 2B was widely spread in 2008 in Northern (Algeria), Central (Cameroon), Eastern (Madagascar, Mauritius, Seychelles, Kenya), Western (Ivory Coast, Ghana, Senegal), and Southern Africa (South Africa), clustering according to country of sampling [41,84,88,89,105]. Clade 2B H1N1 strains sporadically spread in Kenya and Madagascar in 2009 [88,105] (Figure 3, Appendix A).

### 3.4. Molecular Epidemiology of Pandemic H1N1pdm09 Viruses in Africa

We refer to H1N1pdm09 virus strains sampled from 2009 to 2010 and from 2011 onwards as pandemic H1N1pdm09 (pH1N1pdm09) and seasonal H1N1pdm09 (sH1N1pdm09), respectively.

#### 3.4.1. Genomic Diversity of H1N1pdm09 Viruses and Their Relatedness to Vaccines

Africa pH1N1pdm09 virus strains showed early evolution from the A/California/07/2009(H1N1) vaccine strain recommended for global use during the 2010–2016 seasons. The complete H1 gene and protein sequences of the pH1N1pdm09 strains were 97–99.5% and 98.4–99.5% similar to the A/California/07/2009(H1N1) vaccine strain [46,47,48,52,53,54,55,62,65,93,104]. Partial H1 gene sequences of H1N1pdm09 strains sampled from the Central African Republic (CAR) in 2010 [59] and South Africa (Cape Town) in April–July 2011 [65] were 99–100% and 96.6–98.5% similar to the A/California/07/2009(H1N1) strain, respectively.

H1N1pdm09 strains sampled between 2009 and 2011 in Kenya [54,55,93], Tunisia [50], and Morocco [52] had several substitutions at four major antigenic sites: Sa (N125D, S162I, S183P, K164F), Sb (S185T, S190G, S202T), Ca_1_ (H138Q, G170R, S203T, R205K, S220T, D238E), and Ca_2_ (D222G/E, D239E) relative to the A/California/07/2009(H1N1) strain. Despite the rapid genetic diversification, pH1N1pdm09 and early post-pandemic H1N1pdm09 strains (sampled in 2011) remained antigenically similar to A/California/07/2009(H1N1) vaccine strain. Except for two South Africa strains sampled in 2010, with HA1 substitutions N125D and V272F, that reacted low to A/California/07/2009(H1N1) antisera [66].

Africa sH1N1pdm09 strains progressively drifted from the A/California/07/2009(H1N1) vaccine strain, with new substitutions observed every year. H1 proteins of sH1N1pdm09 strains sampled in the Democratic Republic of Congo (Congo), Ethiopia, Egypt, Ghana, Mozambique, Cameroon, and Kenya between 2013 and 2016 had all five major antigenic sites: Sa (T120A, A141V), Sb (S203T, E374K, S451N, S162R/N, S185T), Ca_1_ (D97N, K283E), Ca_2_ (S203T, K163Q), and Cb (A48S, P83S, S84N, K163Q) mutated relative to the A/California/07/2009(H1N1) vaccine strain [58,61,97,102]. sH1N1pdm09 strains sampled between 2017 and 2018 evolved and were 97.9–99.4% and 97.5–99% similar to the new 2017–2018 season A/Michigan/45/2015(H1N1) vaccine strain at the nucleotide and amino acid level, respectively [91]. Kenya H1N1pdm09 viruses sampled during the 2017–2018 season had several mutated antigenic sites: Sa (T120A, P137S, A141E, I149L), Ca_1_ (H273Y, I295V), and Ca_2_ (S164T), Cb (S74R, S162N) relative to A/Michigan/45/2015(H1N1) strain [61]. In addition, Kenya strains sampled in 2018 had mutated antigenic sites, Ca_1_ (G45R, V298I) and Ca_2_ (R223Q), relative to the new 2019 NH season A/Brisbane/02/2018(H1N1) vaccine strain [61]. Both antigenic drift and variation in the dominant antigenic sites between seasons, 2009–2010 (Ca_1_) to 2014–2016 (Cb, Sa, Sb), and 2017–2018 (Ca_1_, Cb) resulted in low efficacy of the A/California/07/2009(H1N1), A/Michigan/45/2015(H1N1), and A/Brisbane/02/2018(H1N1) vaccine strains of 24.55–35.77%, 39.6–41.8%, and 32.4–42.1% against the 2014–2016 and 2017–2018, and 2018 sH1N1pdm09 viruses, respectively [58,61].

All six classical N-linked glycosylation sites present in the A/California/07/2009(H1N1) strain HA1 protein were conserved among 2009–2011 and 2014–2016 H1N1pdm09 strains sampled in Tunisia [50] and Cameroon [58], respectively. Except for A/Tunisia/15656/2009(H1N1) and A/Tunisia/2144/2011(H1N1) strains with substitutions N276H and T288N, respectively. Kenya strains sampled during the 2015–2016 season retained four (11, 23, 87, and 287) of the six classical N-linked glycosylation sites. While all 2015–2017 and 2017–2018 sH1N1pdm09 strains sampled in Egypt [101] and Kenya [61], respectively, gained a new N-linked glycosylation site (S162N) present in the A/Michigan/45/2015(H1N1) strain H1 protein, except for one A/Kenya/066/2018(H1N1) strain.

Africa H1N1pdm09 strains also evolved from vaccine strains in the remaining seven genes (PB2, PB1, PA, NP, N1, MP, and NS). For example, Kenya H1N1pdm09 strains sampled in 2009 had PB2, PB1, and PA proteins with 99–100%, 98.4–100%, and 99.4–100% similarity to A/California/07/2009(H1N1) strain [55]. All pH1N1pdm09 strains sampled in Uganda, Senegal, Mauritania, Kenya, and South Africa had V106I and N248D in the catalytic site (CS) of their N1 proteins relative to the A/California/07/2009(H1N1) strain [47,48,54,55,56,104]. Quiliano et al. observed 26 unique substitutions among N1 proteins of 59 Africa H1N1pdm09 strains sampled between 2009 and April 2011, with the highest frequency of substitutions in the transmembrane (TM) and L21 domains [64]. Kenya pH1N1pdm09 strain nucleoproteins (NP) and matrix protein (M1 and M2) sequences sampled in 2009 were 98.8–99.6% and 98–100%, similar to the A/California/07/2009(H1N1) strain, respectively. M1 (92–100%) was more conserved than M2 (77–100%) proteins among Kenya pH1N1pdm09 strains [54]. Virus strains sampled from Kenya, Nigeria, Ethiopia, and Mali in 2009 and Senegal in 2010 had NS proteins with a mean amino acid similarity of 98% to A/California/07/2009(H1N1) strain [53]. NS1 was more conserved than NS2 proteins among Kenya pH1N1pdm09 strains [53,54,55].

#### 3.4.2. Genomic Markers of Disease Severity among H1N1pdm09 Viruses

H1N1pdm09 strains with antigenic site Ca_2_ substitution, D222G, associated with disease severity [106], circulated at lower frequencies in Tunisia (8%, 4/50) and Egypt (9%, 1/17) during the 2009–2011 and 2009 season, respectively [49,50,101]. Another D222G variant was sampled in Egypt later in 2016 [101]. Several HA1 substitutions, S185T, Q293H, K374E, 312I, H138Q, S203T, R205K, D222E, and T288N, were also sporadically observed among patients with severe acute respiratory illnesses (SARI) [49,50,51,66]. The D222E variants widely circulated in Tunisia, South Africa, Morocco, Senegal, and Egypt during the pandemic [48,49,50,51,52,66,101].

#### 3.4.3. Antiviral Drugs Sensitivity among H1N1pdm09 Viruses

All Africa H1N1pdm09 virus strains sampled between 2009 and 2016 had the M2 adamantane-resistant marker, S31N [47,54,55,58,105], but lacked the N1 oseltamivir-resistance marker, H275Y [47,48,51,54,55,58,62,63,66,90,101,104]. Except for a strain sampled from a South African patient after treatment with a standard dose of oseltamivir during the pandemic [43] and an A/Egypt/1424/2016(H1N1) strain had H275Y in their N1 proteins. No H1N1pdm09-H275Y variant was sampled in 2017. All N1 sites, 117, 119, 136, 151, 152, 199, 223, 247, 293, and 295, associated with reduced NAIs inhibition in vitro were conserved among Cameroon 2014–2016 sH1N1pdm09 strains [58].

#### 3.4.4. Phylogenetic Clustering and Circulating Clades among H1N1pdm09 Viruses

Phylogenies showed that Africa H1N1pdm09 virus strains were variants of the A/California/07/2009(H1N1) strain that diverged into multiple clades that co-circulated every season. Analysis of concatenated whole viral genomes (concat-8) formed similar but deeper phylogenies (with a higher number of nodes between root and furthest tip) than those of individual genes [54,55]. Pandemic H1N1pdm09 viruses sampled from different African countries mixed. While sH1N1pdm09 strains clustered distinctly according to time and location (household or province or country) of sampling [47,53,55,60,62,102,103].

The 2009–2011 H1N1pdm09 outbreaks observed in different African countries resulted from the emergence of strains belonging to genetic clades 2, 3, 5, 6, and 7 (Figure 4). Clade 2 pH1N1pdm09 strains, similar to earlier Mexican- and England-origin strains sampled in April–June 2009, were first introduced in Kenya on 19 April 2009 (95% CI: 27 December 2008–6 July 2009) [54,55]. Similar clade 2-like strains with HA1 substitutions N31D and S162N similar to A/Mexico/2466/07/2009(H1N1), A/Wisconsin/02/01/2011(H1N1), and A/Czech Republic/32/2011(H1N1) strains were sampled in South Africa later in 2010 [50,51,52,66]. Novel clade 3 strains with HA1 substitutions S183P and A134T, similar to the A/England/195/2009(H1N1) and A/New York/3177/2009(H1N1) strains, were first sampled in Cameroon in 2009 and circulated through 2010 [48]. Similar clade 3 strains were also observed in Ghana, Senegal, Ivory Coast, and Ethiopia in 2010 and last reported in Tunisia in 2011 [48,50,51,52,54,55,91]. Pandemic clade 5 strains, with HA1 substitutions D97N, R205K, I216V, and V249L, first emerged in 2010 in Morocco [52], then continued to circulate in Morocco, Tunisia, South Africa, and Uganda in 2011 [50,51,52,95,101]. Shockingly, clade 5 remerged in Tunisia later in 2017 [91].

Novel clade 7 with HA1 substitutions S185T, S143G, and A197T were first sampled in 2009 in Egypt, Nigeria, Reunion, Mauritius, Tanzania, and Kenya [48,54,55,63,98] (Figure 4). Molecular dating of the H1 phylogeny estimated that clade 7 strains, similar to earlier strains sampled from England, California, and China, were introduced in Kenya (KENB-GC7) on 4 June 2009 (95% CI: 16 April–29 June 2009), 2 months before laboratory detection [54,55] and in the Reunion (RUN) on 19 May 2009 [63]. Contrastingly, whole-genome (concat-8) analysis estimated the most recent common ancestor (TMRCA) of Kenya pH1N1pdm09 strains at an earlier date of 28 February 2009 (95% CI: October 2008–May 2009) [54,55]. Notably, clade 7 diversified in Kenya through independent introductions of multiple subclades: KENC-GC7, KEND-GC7, KENE-GC7, and KENF-GC7 introduced on 8 August 2009 (19 May–2 October 2009), 7 August 2009 (10 June–17 September 2009), 4 August 2009 (10 June–2 October 2009), and 2 October 2009 (20 June–10 December 2009), respectively [54,55]. Similarly, two subclades of clade 7 (RUNA and RUNB), similar to a Japan-origin A/Sapporo/1/2009(H1N1) strain, were introduced in Reunion on the 26th June 2009 and 8th July 2010, respectively [63]. Clade 7 strains sampled in Reunion, Tanzania, and Mauritius were more similar than those sampled in Seychelles and Madagascar. Clade 7 continued to circulate in 2010 (Senegal, Kenya, Mauritania, South Africa, Cameroon, and Nigeria), 2011 (Kenya and Uganda), and 2012 (Tanzania, Kenya, and Madagascar) [48,54,55,84,96].

Another group of the earliest pH1N1pdm09 strains sampled in Africa belonged to clade 6, with HA1 substitutions D97N and S185T, similar to A/New York/3324/2009(H1N1) and A/Shanghai/143T/2009(H1N1), which later became dominant and the most diverse. Clade 6 strains were first sampled in Cape Verde and Kenya in 2009 [48,98]. Clade 6 continued to circulate in Kenya, Morocco, Ghana, South Africa during the 2010–2011 season and in Tunisia (2011 and 2013) [51,52,92,93,98,105]. After the 2009–2010 pandemic, several clade 6 variants: 6A, 6B, 6B.1, 6B.1a, 6B.2, 6B.3, and 6C emerged in different African countries. Specifically, novel clade 6C strains, with HA1 substitutions D97N, S185T, K283E, E499K, and V234I, were first sampled in Ivory Coast in 2012 [99], then circulated in different Western and Eastern African countries between 2013 and 2014 [58,94,96,97,98,99,101,102] (Figure 4 and Appendix A). Novel sH1N1pdm09 viruses belonging to clade 6B, with HA1 substitutions K163Q and A256T, emerged in 2013 in Ivory Coast, Nigeria, and South Africa [58,94,97,99,102], before it widely spread in all African regions from 2014 through 2016 [58,94,96,97,98,99,102]. Among the clade 6B variants, 6B.3 strains were sampled only in Egypt during the 2013–2014 season [101]. In contrast, 6B.2 with HA1 substitutions E499K, S84N, and R45K and 6B.1 with S84N, S162N, and I216T circulated in the 2014–2016 and 2015–2018 seasons, respectively, in different countries [58,61,91,96,98,99,101,102]. Clade 6B.1 dominated and further diverged into 6B.1a and 6B.1a1 subclades, which were observed in 2018 in Kenya [98]. A study by Opanda et al. reported the emergence of novel A/Brisbane/02/2018(H1N1)-like clade 6B.1a1 strains in Kenya in 2018 [61]. Strains with a clade 6A-like H1 genes were sampled in 2014 and 2016 only in Uganda [58].

Three novel clades, A/Madrid/SO8171/2010(H1N1)-like with E172K, K308E, and V47I, A/Christchurch/16/2010(H1N1)-like (clade 4) with N125D, and A/Cameroon/LEID/01/11/1450/2011(H1N1)-like (clade 8, unique to Africa) with A186T and V272A substitutions in their HA1 proteins emerged in the early post-pandemic year (2011) circulating in Tunisia, Morocco, and three countries (Ghana, Nigeria, and Cameroon), respectively [50,51,52,95]. Like clade 8, novel clade 9 with HA1 substitutions L32I, D86E, S128T, R259K, S263A, I460V, and V520A was unique to Africa (Appendix A), observed in Senegal and Ghana in 2012 and 2013, respectively [97,102].

Despite the multiple introductions and early diversification of H1N1pdm09 strains, two highly supported viral clusters (I and II, bootstrap > 80%) persisted for >1.8 years in Western Africa between 2010 and 2012 [60]. Cluster I contained strains with H1 substitutions S145T and R276K that circulated in Senegal, Ghana, and Ivory Coast between March 2011 and January 2013, similar to a France-origin strain sampled earlier in 2010. The largest cluster II contained both Western Africa (*n* = 26, including the A/Ghana/763/2011(H1N1)-clade 8 strain) and non-Western Africa (Ethiopia, France, Italy, Norway, Sweden, and Minnesota, *n* = 11) strains with H1 substitutions A15T, N490D, T491K, and V537, sampled during the November 2010–October 2012 and November 2012–January 2013 seasons, respectively. All cluster II non-Western Africa strains clustered monophyletically (bootstrap = 100%) [60].

Phylogeographic analysis by Owuor using the Bayesian Ancestral state reconstruction (BSSVS) confirmed the introduction of H1N1pdm09 strains into Africa between 2009 and 2018 from outside Africa, with the highest viral migration rate of 1.48 (Bayes factor, BF ≥ 3) from East and Southeast Asia [98]. Introduction events were inferred as clusters with bootstrap ≥ 80%, size of *n* ≥ 2 strains, and >80% of strains sampled from Africa. The BSSVS model estimated strongly supported H1N1pdm09 viral migration from Northern to Southern Africa (BF ≥ 1000), between countries (100 ≤ BF < 1000), and within the country (Kenya) (rate = 0.81–0.93, BF ≥ 1000) [98]. Overall, H1N1pdm09 strains sampled in Africa belonged to similar globally recognized clades. However, most clades first circulated elsewhere before their introduction in Africa, except clades 8 and 9 were unique to Africa (Appendix A).

#### 3.4.5. Population Dynamics, Evolutionary Rates, Selection, and Reassortment among H1N1pdm09 Viruses

Upon viral introduction in Kenya, the population sizes (genomic diversity) of H1N1pdm09 strains fluctuated continuously, with the highest peaks observed during the 2009–2010 and 2018 seasons [98].

Using a relaxed molecular clock, concatenated whole genomes of Kenya pH1N1pdm09 strains were estimated to evolve at a mean rate of 4.9 × 10^−3^ (2.6–7.2) substitutions per site per year (subs/site/year). While their individual genes evolved at different mean rates such as MP at 9.88 × 10^−3^ (5.58–14.5), H1 at 5.58 × 10^−3^ (2.75–9.28), NS at 5.22 × 10^−3^ (1.64–9.17), N1 at 4.07 × 10^−3^ (1.47–7.73), PB2 at 4.01 × 10^−3^ (1.47–6.45), PB1 at 3.89 × 10^−3^ (1.29–7.22), PA at 1.86 × 10^−3^ (3.03–6.19), and NP at 0.80 × 10^−3^ (0.4–2.04) subs/site/year [54,55]. Venter et al. applied a strict clock on the HA1 genes of South Africa pH1N1pdm09 strains and estimated a lower mean rate of 0.9 × 10^−4^ subs/site/year [66]. All South Africa and Kenya H1N1pdm09 strains sampled during the 2009–2010 and 2015–2018 seasons, respectively, exhibited lower rates of nonsynonymous (dN) than synonymous substitutions (dS) with dN/dS ratio of 0.6–0.8 in their HA1 genes, confirming purifying (negative) selection [61,66]. Single likelihood ancestor counting (SLAC) and fixed-effects likelihood (FEL) predicted no codon site under selection at a *p*-value <0.05 [61].

Gachara et al. [55] did not predict any reassortant among Kenya pH1N1pdm09 strains using the FluGenome web tool [107]. However, H1N1pdm09 strains sampled from 2009 to 2016 across Africa shared similar H1 but different N1 genes, suggesting a different origin of N1 genes [58,60]. Furthermore, two Uganda strains (A/Uganda/856/2014(H1N1) and A/Uganda/1744/2016(H1N1)) had clade 6A-like H1 genes, but clade 6C- and 6B-like N1 genes, respectively, suggesting reassortment [58].

### 3.5. Molecular Epidemiology of Seasonal H3N2 Viruses in Africa

#### 3.5.1. Genomic Diversity of H3N2 Viruses and Their Relatedness to Vaccines

Africa seasonal H3N2 strains showed an early and continuous evolution from vaccine strains every year. Three South Africa H3N2 strains sampled in 1993 had eight to nine substitutions at three major antigenic sites A, B, and C relative to the season’s A/Beijing/353/1989(H3N2) vaccine strain [68]. South Africa strains sampled in 1994 evolved at four major antigenic sites, A, B, C, and D, and were genetically and antigenically similar to a new A/Guangdong/25/1993(H3N2) strain [68]. Besselaar et al. sampled an A/South Africa/1147/1996(H3N2) strain with an HA1 protein similar to the new A/Nanchang/933/1995(H3N2) vaccine strain [69]. A/Nanchang/933/1995(H3N2)-like strains widely circulated later in 1997 in South Africa, Senegal, and Morocco [69,83,89]. H3N2 strains sampled in February 1998 (early) and later in 1998 had HA1 proteins similar to the different vaccine strains, A/Nanchang/933/1995(H3N2) and new A/Sydney/5/1997(H3N2), respectively [69,83,85]. Hemagglutinin-inhibition confirmed that all and the majority of H3N2 strains sampled in 1997 and 1998, respectively, reacted to antisera raised against A/Nanchang/933/1995(H3N2) and A/Sydney/5/1997(H3N2) strains, respectively, except for A/Jhb/1/1998(H3N2) and A/Jhb/2/1998(H3N2) strains [69,83,85]. Two new viral lineages of Europe-origin (A/Finland/620/1999(H3N2)) and Central America-origin (A/Panama/2007/1999(H3N2)) emerged in South Africa in 1999 [85]. A/Panama/2007/1999-like strains circulated in Senegal and South Africa from 1999 to 2002 [70,89] and caused the 2002 outbreak in Bosobolo (Congo) [82].

The following year (2003), novel A/Fujian/411/2002(H3N2)-like strains with HA1 substitutions A131T and H155T emerged in Senegal and caused an outbreak in a police residential college in Pretoria, South Africa [70,89]. The Pretoria outbreak strains lacked an HA1 Q156H present in strains sampled the same year from Johannesburg and Middleburg (South Africa). Dakar (Senegal) H3N2 strains sampled in 2004 were genetically similar to the 2005 SH season vaccine strain, A/Wellington/1/2004(H3N2) [89]. Four strains sampled in Nairobi (Kenya) between July and September 2006 were antigenically similar to the 2006 SH season vaccine variants, A/California/07/2004(H3N2) and A/New York/55/2004(H3N2) [71]. Kenya H3N2 strains sampled in 2007 had mutated major antigenic sites A (G50E, D122N, S138A, K140I), B (V186G), and I223V relative to the 2007 SH season A/Wisconsin/67/2005(H3N2) vaccine strain but remained antigenically similar to the vaccine [71,80]. A total of 50 Uganda H3N2 strains sampled between October and December 2008 had all 8 genes similar to the 2007–2008 NH A/Wisconsin/67/2005(H3N2) and 2008–2009 SH A/Brisbane/10/2007(H3N2) vaccine strains [75].

New A/Perth/16/2009(H3N2)-like strains emerged and co-circulated with pH1N1pdm09 strains in Uganda and Senegal in 2009 [73,89]. The Uganda H3N2 strains sampled in 2009 differed from the 2010–2012 SH season A/Perth/16/2009(H3N2) vaccine strain at antigenic sites B (N189K, N144S, K158N) and D (T212A) [73]. More mutated antigenic sites E (K62E), C (S45N), B (L157S), A (K144N), and E (G78D) were observed among Kenya strains sampled in the 2010–2011 season relative to the A/Perth/16/2009(H3N2) strain [80,93]. Kenya H3N2 strains sampled in 2013 differed from the 2013 SH season A/Victoria/361/2011(H3N2) vaccine strain, at mutated antigenic sites A (I140R, I140G, N145S), B (T128A, R156H, V186G), C (N278K), D (G78D), Q33R, R142G, E190D, and Y219S [80]. The dominant antigenic sites varied among Kenya strains in 2007 (A: D122N, S138A, K140I), 2008 (A: N144S/K), 2009 (B: K158N and N189K), 2010 (A: N144K), and 2013 (A: T128A, R156H, and V186G), resulting in a vaccine efficacy of 17%, 72.36%, 49.98%, 72.36%, and 17%, respectively.

In 2014, new strains similar to the 2014 SH season A/Texas/50/2012(H3N2) vaccine strain were sampled in Congo [97], South Africa [77], and Southern and Northern Cameroon [78,79]. The A/Texas/50/2012(H3N2) vaccine showed low efficacy in South Africa (VE: −18.4% (−171.5 to −48.4) [77], and the majority of the sampled Congo and South Cameroon strains were also antigenically similar to the A/Switzerland/9715293/2013(H3N2) vaccine strain used later in 2015 [78,97]. A total of 50% (6/12) of Egypt H3N2 strains sampled in October–December 2014 had substitution N225D in H3 and N151D in N2 associated with non-agglutination of avian red blood cells (RBCs) [67]. H3N2 strains sampled in Egypt strains between 2015 and 2017 had several mutated antigenic sites A (T135K, R142G/K/A, N144K/S, N145S), B (L157S, F159Y, K160T), C (D53N/G, N278K), and D (N121D/K/S, N225D), and all had the non-agglutinating substitution N225D [101]. Strains sampled in the same period (2015–2017) from Northern and Southern Cameroon were similar to the 2016–2017 season SNH A/Hong Kong/4801/2014(H3N2) vaccine strain but had different dominant antigenic sites resulting in different efficacy of the A/Switzerland/9715293/2013(H3N2) vaccine in Northern (80.55%) and Southern Cameroon (17%) in 2015 [78,79].

Africa H3N2 strains also evolved from vaccine strains by altering potential N-glycosylation sites (pNGS) in their H3 proteins. Specifically, 100% and 89.5% of Kenya H3N2 strains sampled in the 2010 and 2010–2011 season gained new pNGS K144N and S45N, respectively, absent in the A/Perth/16/2009(H3N2) vaccine strain [93]. Another substitution N144D resulted in a loss of a pNGS among strains sampled in Ghana, Nigeria, and South Africa in 2011, and one A/Johannesburg/27/2011(H3N2) strain gained a pNGS due to S45N [95]. A total of 5 Tunisia H3N2 strains sampled in January–February 2013 had 1 mutated receptor-binding site (S145N) and 10 substitutions at 5 pNGS (45, 124, 128, 144, and 145) relative to the A/Perth/16/2009(H3N2) strain [74]. A total of 60% (6/10) of the pNGS substitutions were among severe and fatal cases [74]. Two strains, A/Tunisia/2494/2013(H3N2)-fatal and A/Tunisia/1987/2013(H3N2)-severe, had an additional mutated receptor-binding site (RBS) A198S [74]. Contrastingly, Nyang’au et al. predicted six HA1 pNGS (8, 22, 63, 133, 165, and 285) as conserved (threshold > 0.5) among 115 Kenya strains sampled from 2007 to 2013 [80]. South Africa and Cameroon H3N2 strains sampled during the 2014–2016 seasons had varying numbers of pNGS (8–11) [77,78,79]. However, the number of pNGS did not affect viral reaction to the A/Texas/50/2012(H3N2) vaccine antisera among South Africa strains sampled in 2014 (*p*-value = 0.6236) [77]. Cameroon strains sampled in the 2014–2016 seasons had pNGS at H3 positions (8, 22, 38, 45, 63, 128, 133, 160, 165, 246, and 285). Substitutions A128T and K160T created new pNGSs [78,79]. Virus strains sampled in 2015 from Mozambique lost two (L3I, N144S) and gained three (F159Y, K160T, N225D) pNGSs [102], and the Gambia lost one (N144S) and gained one (K160T) pNGSs [96] in their H3 proteins. A total of 94.7% (71/75) of strains sampled in Kilifi (coastal Kenya) between December 2015 and March 2017 gained a new pNGS K160T, but their RBS (98, 136, 153, 183, 190, 194, 195, and 228) remained conserved [81].

All Uganda H3N2 strains sampled during the 2008–2009 season had the classical receptor-binding site motif with 19Y, 136S, 153W, 183H, 195Y, and 225–228 NIPS conserved [73].

South Africa and Cameroon H3N2 strains sampled between 2009 and 2016 also continuously evolved from vaccine strains in their N2 and MP genes [78,104].

#### 3.5.2. Antiviral Drugs Sensitivity among H3N2 Viruses

Six H3N2 strains sampled from Egypt and South Africa during the 2004–2006 seasons lacked the primary adamantane-resistance marker S31N in their M2 proteins [87]. Adamantane-resistance variants (with S31N) emerged in 2008 and continued to circulate in Uganda, Kenya, and Cameroon during the 2008–2009, 2008–2011, and 2014–2016 seasons, respectively [72,73,78,79,105]. H3N2-S31N variants sampled in Kenya, Malaysia, and Singapore between 2008 and 2011 shared HA1 substitutions F193S and I140K [105]. A/KEN/164/2011(H3N2) [105] and A/Cameroon/16v-6529/2016(H3N2) strains [79] had four (L26I, V27S, A30C, and S31D) and one (V27A) secondary M2 adamantane-resistant markers, respectively.

All H3N2 strains sampled in Africa between 2007 and 2016 lacked the oseltamivir-resistance marker H275Y in their N2 proteins [73,78,79,90,104]. Neuraminidase inhibition (NAI) assays confirmed that all strains sampled in Cameroon, Ivory Coast, Madagascar, Niger, Senegal, Congo, and Mozambique during the 2008–2010 and 2014–2015 seasons were sensitive to oseltamivir and zanamivir drugs [88,102].

#### 3.5.3. Phylogenetic Clustering and Circulating Clades among H3N2 Viruses

Phylogenetic trees of Africa H3N2 strains showed the cocirculation of multiple viral lineages, temporal evolution, and mixture with global strains [26,73,83,85,89]. The earliest strains sampled from South Africa in 1994 (A/Johannesburg/33/1994(H3N2), A/Johannesburg/47/1994(H3N2)) and 1997 (A/Johannesburg/10/1997(H3N2)) clustered with Asian-origin A/Beijing/32/1992(H3N2) and A/Wuhan/359/1995(H3N2)-like strains, respectively [26]. Using an automated Graph-incompatibility-based Reassortment Finder (GiRaF) tool, Westgeest et al. predicted the 3 Johannesburg strains had reassorted between H3 and PB1, NP, N2, and MP genes [26]. South Africa strains sampled in late 1997, 1998, 1999, and December 1999–2002 seasons clustered with earlier strains sampled from Asia and Europe(A/Guangdong/8/1996(H3N2), A/Bratislava/6/1997(H3N2)), Australia (A/Sydney/5/1997(H3N2)), Europe (A/Finland/620/1999), and Central America (A/Panama/2007/1999), respectively [70,85].

During the 2003 season, novel strains with HA1 substitution Q156H belonging to the A/Fujian/411/2002(H3N2) clade emerged in South Africa, Senegal, and Madagascar [70,71,89] (Figure 5). The following year (2014), A/Fujian/411/2002(H3N2) clade was replaced by novel A/Wellington/1/2004(H3N2)-like strains in Senegal and South Africa [71,89]. No study reported clades for H3N2 strains sampled in 2005. However, Deyde et al. sampled an A/Egypt/4864/2005(H3N2) strain that had HA1 substitution P227S, similar to an earlier A/Indonesia/4027/2004 strain [87].

The H3N2 epidemics in 2006 (Ghana and Morocco) and 2007 (Cameroon, Kenya, and South Africa) were caused by strains belonging to a novel A/Brisbane/10/2007(H3N2)-like clade with HA1 substitution K140I [70,71,80]. A/Brisbane/10/2007(H3N2)-like strains sampled in Kenya and Uganda later in 2008 acquired new HA1 substitutions I140K, V112I, and F193S [73,80,105]. H3N2 strains with HA1 substitution V112I and an additional K158R were also observed in Cameroon and Senegal in May–August 2009 [88]. Novel A/Perth/16/2009(H3N2) clade strains, with HA1 substitution E62K, emerged in Madagascar in 2008 [88].

During the H1N1pdm09 pandemic year (2009), H3N2 strains belonging to three clades A/Brisbane/10/2007(H3N2), A/Perth/16/2009(H3N2), and A/Victoria/208/2009(H3N2) co-circulated in different Africa countries (Figure 5 and Appendix A). Specifically, A/Brisbane/10/2007(H3N2)-like strains, with K173Q, were sampled in Ghana, Egypt, Kenya, South Africa, Algeria, Madagascar, Tunisia, Ivory Coast, Nigeria, and Cameroon [73,80]. Furthermore, A/Perth/16/2009(H3N2)-like strains, with more characteristic substitutions (E62K, N144K, K158N, N189K, and I230V) circulated in Madagascar, Niger, Senegal, Uganda, Ghana, and Egypt in 2009 [73,80,88,89,105], Niger, Kenya, and Ghana in 2010 [88,95,105], and in Kenya in 2011 [105]. The novel A/Victoria/208/2009(H3N2)-like strains with HA1 substitutions K158N, N189K, T212A, and N312S, emerged and circulated in different countries across all regions from 2009 through 2010 [73,80,81,88]. The earliest variants of the A/Victoria/208/2009(H3N2) clade to be observed belonged to clades 4 and 7, which emerged in Egypt in 2009 [101] and in Kenya in 2010, respectively [81]. Clade 7 strains with HA1 substitutions S45N were later sampled in South Africa, Tanzania, and Kenya in 2011 and in South Africa and Kenya in 2012 [81,95].

During the early post-pandemic year (2011), new A/Victoria/208/2009(H3N2) clade variants, clades 5, 3 or 3A, 3B, and 3C, emerged in Africa. Novel clade 5 strains with HA1 substitutions D53N, Y94H, I230V, and E280A, similar to the A/Alabama/05/2010(H3N2) strain, were only sampled in 2011 in Madagascar and Reunion [95]. Novel European-origin (A/Stockholm/18/2011(H3N2)-like) clade 3 (same as 3A) with HA1 substitutions N145S, V223I, and N144D, were also first sampled in 2011 in Ghana, Nigeria, and South Africa [95], and later sampled South Africa and Ethiopia in 2012 [92,101] and in Tunisia during the 2012–2013 season [91]. Novel clade 3B strains with HA1 substitutions N312S and A198S were first sampled in 2011 in Kenya, Tunisia, and South Africa, then in Tunisia, Egypt, Morocco, Madagascar in 2012 [80,81,92]. The novel Asian-origin A/Hong Kong/3969/2011(H3N2)-like clade 3C strains with HA1 substitutions S45N, T48I, A198S, and V223I were first sampled in 2011 in South Africa [80,95].

H3N2 virus epidemics that occurred in Africa in 2012 onwards were dominated by clade 3C and its variants (3C.1, 3C.2, 3C.2a, 3C.2a1, 3C.2a1b, 3C.2a2, 3C.2a3, 3C.2a4, 3C.3, 3C.3a, and 3C.3b) (Figure 5). The classical clade 3C strains continued to be sampled during 2012 (in Senegal and Kenya), 2012–2013 (in Mauritius), and 2014–2015 (in Kenya) seasons [80,81,92]. The earliest clade 3C variants belonged to clade 3C.3 and were sampled in Algeria, the Gambia, and Ivory Coast in 2012 [99,101]. Notably, multiple clade 3C variants co-circulated each year from 2013 through 2018. For example, novel clade 3C.1 strains with HA1 substitutions Q33R, N145S, and N278K emerged in Tunisia and Ethiopia in 2013 [80]. In the same year (2013), novel A/Hong Kong/146/2013(H3N2)-like clade 3C.2 strains with N145S and A/Samara/73/2013(H3N2)-like clade 3C.3 strains with T128A and R142G were sampled in Tunisia [91] and different countries (Nigeria, Ivory Coast, South Africa, Ghana, Kenya, Cameroon, and Togo) [80,99,101], respectively. Clades 3C.3 and 3C.2 continued to circulate in different countries during the 2014–2016 [78,97,99] and 2015–2017 [78,101] seasons, respectively.

During the 2014 season, clade 3C.3 diverged into 3C.3a and 3C.3b. Clade 3C.3b strains had HA1 substitutions K83R and R261Q and were sampled in 2014 only in Egypt [97]. While clade 3C.3a strains with HA1 substitutions N225D (some with A138S, F159S, Y94H, or K326R) widely spread in Burkina Faso, Ghana, Senegal, South Africa, Ethiopia, Tanzania, Madagascar, Cameroon, and Nigeria [77,78,79,96,97,99]. Clade 3C.3a strains continued to circulate in Nigeria and Cameroon the following year (2015) and were last observed in 2016 in Uganda [79,96,99].

Novel clade 3C.2a, with HA1 substitutions L3I, K160T, N225D, F159Y, and N128K, dominated the 2014–2017 seasons, circulating in 65% (22/34) of the countries sampled [76,77,79,81,96,97,98,99,102]. However, there were notable differences in the patterns of clade circulation within countries. For example, both clade 3C.2a and 3C.3a strains co-circulated in Northern and Southern Cameroon during the 2014–2015 seasons, but 3C.3a strains emerged in Northern in 2015, later than in Southern Cameroon [78,79]. In addition, subclade 3C.3a circulated in Burkina Faso from May to mid-September 2014, but 3C.2a emerged later in August–October 2015 [99]. Furthermore, H3N2 strains sampled in Cameroon, Tanzania, Ethiopia, and Nigeria between 2014 and 2016 switched between clades 3C.3, 3C.2a, and 3C.3a in their N2 and MP phylogenies suggestive of reassortment [78,79].

Phylogenetic analysis revealed a highly supported (bootstrap = 99%) transmission of novel subclade 3C.2a1 viruses with N171K between two South African provinces. Specifically, a strain sampled in the Western Cape Province A/South Africa/3743/2016(H3N2) was introduced and resulted in an outbreak in a boarding school in the Eastern Cape Province between 13 and 29 July 2016 [76]. In 2016, clade 3C.2a diverged into novel subclades 3C.2a1, 3C.2a1b, 3C.2a2, and 3C.2a3 and all four variants co-circulated in 2017 in different countries. Specifically, clade 3C.2a2 strains with T131K, R142K, and R261Q were only sampled in Mauritius (2018) and Zambia (2017) [79]. A study by Owuor et al. observed both 3C.2a1b and 3C.2a3 strains in Kenya in 2017 [81]. Another novel subclade 3C.2a4 with HA1 substitutions N31S, D53N, N162K, and I192T emerged in Tunisia later in 2017 [79]. Subclades 3C.2a1 strains with HA1 substitutions dominated the 2017 season in Ethiopia, Morocco, Tunisia, Congo, Tanzania, and Kenya, and the 2018 season in Egypt, Madagascar, and Mauritius [79] (Figure 5 and Appendix A).

Similar to H1N1pdm09, we observed differences in circulating H3N2 clades between Africa and elsewhere in the world since 2009 (Appendix A).

#### 3.5.4. Viral Population Dynamics, Evolutionary Rates, Selection, and Phylogeographic Patterns among H3N2 Viruses

Bayesian phylogenetics and molecular dating confirmed multiple introductions of new H3N2 strains in Africa from elsewhere in the world. Nyang’au et al. applied a relaxed clock on the HA1 phylogeny of Kenya H3N2 strains sampled between 2007 and 2013 and dated their most recent common ancestor (TMRCA) on September 2001 (95% high probability density (HPD) = September 1998–October 2003) [80]. Molecular dating of concatenated whole genomes of Kenya strains sampled during the 2015–2016 season estimated different TMRCA ages for clades 3C.2a, 3C.2a1b, and 3C.2a2 on 12 January 2013, 26 June 2014, and 2 May 2014, respectively [98]. The HA1 genes of the 2007–2013 season Kenya H3N2 strains evolved at a mean rate of 4.17 × 10^−3^ (3.09–5.31) subs/site/year and underwent negative selection (dN/dS = 0.56). None of their codons was under selection [80].

Lemey et al. did Bayesian analysis on 1529 global H3N2 virus sequences (including Africa, *n* = 31) sampled between 2002 and 2007, human morbidity, and air transportation data. Lemey et al. identified South Africa as the only and largest air community connecting Eastern, Western, and Southern Africa [21]. Trunk analysis showed that mainland China and Southeast China contributed 60% and 15% of the strains in the global H3 phylogeny trunk, respectively. Africa did not contribute to the tree trunk [21]. Bayesian ancestral state reconstruction (BSSVs) analysis using concatenated H3N2 whole genomes confirmed highly supported viral migration rates at 0.62–3.34 from East and Southeast Asia to Africa and other continents between January 2010 and December 2013 [98]. Furthermore, H3N2 viral migration within Africa was also highly supported, from Northern to Southern hemispheres (BF ≥ 1000) and within and between geographical regions (10 ≤ BF < 100) of the continent. Dense populations and close locations favored viral migration in Kenya [98].

### 3.6. Reassortment between H1N1pdm09 and H3N2 Viruses and Zoonotic Exchanges

Four Kenya H3N2 strains, A/Kenya/201/2010(H3N2), A/Kenya/206/2010(H3N2), A/Kenya/209/2010(H3N2), and A/Kenya/214/2010(H3N2), sampled between October and December 2010 were identified as reassortants with A/Perth/16/2009(H3N2)-like H3 and A/California/7/2009(H1N1)-like MP genes [72]. Another study later confirmed three of these strains, A/Kenya/201/2010(H3N2), A/Kenya/206/2010(H3N2), and A/Kenya/209/2010(H3N2), to have reassorted M2 proteins with substitutions I11T, N13S, G16E, V28I, and I54R similar to H1N1pdm09 strains sampled in the 2009–2011 season [105].

There was evidence of swine infection with human IAVs and phylogenetic mixing of avian, swine, and human IAVs in Africa. Phylogenetic analysis 102 virus sequences sampled in Tunisia from humans (H1, H3, N1, and N2) and avian (H9, PB2, NP, and MP) between 2009 and 2013 showed clustering of human (H1, H3) and an avian A/chicken/Tunisia/848/2011(H9N2) strain [100]. Soli et al. used the Recombination Detection Program (RDP) software v.4.beta79 [108] and predicated 12 recombination events (*p*-value < 10^−5^) among H3, H1, and H9, with H1 and H9 sequences as major parents in 9 and 2 events, respectively. A total of 67% (8/12) of the events resulted in H3 sequences with recombined sites ≥ 100 base pairs [100].

A total of 14% (31/227) of swine sampled in Nigeria between July 2010 and June 2012 were infected with human H1N1pdm09 strains, and all 31 infections were observed in 2011. The Nigeria swine H1N1pdm09 strains clustered with human H1N1pdm09 virus strains sampled earlier during the 2010–2011 season in Western Africa and globally [57]. A total of 46 human and 4 swine H1N1pdm09 strains sampled during the 2009–2013 and 2010–2013 seasons, respectively, mixed regardless of their country of sampling in their partial H1 (positions 1–138 and 1069–1752 deleted) and N1 (positions 1–598 and 1318–1410 deleted) phylogenies [56,57]. A syndromic survey detected human H1N1pdm09 and H3N2 strains among 9.8% (13/132) and 2.3% (3/132) of Ghanaian swine sampled between 2014 and 2015, respectively. Phylogenetic analysis of the MP genes showed clustering of a human (A/Ghana/KM001/2015(H1N1)) and swine H1N1pdm09 strains (A/swine/Nigeria/IBDVR004/2015(H1N1) and A/swine/Nigeria/IBDVR005/2015) (bootstrap = 68%) [42]. The following season (2016–2017), H1N1pdm09 strains were detected in 1.4% (17/1200) of Ghanaian swine [44]. All swine H1N1pdm09 strains sampled in Nigeria, Ghana, Cameroon, Kenya, Togo, and Egypt between 2014 and 2017 evolved from the human A/California/04/2009(H1N1) vaccine strain and clustered according to the year of sampling in the H1 and MP phylogenies [42,44]. Antigenic analysis found antibodies against human H3N2 and H1N1pdm09 strains in swine and farmers sera, confirming bidirectional viral transfer [44].

## 4. Discussion

We successfully merged data on the genomic evolution, diversity, transmission, and migration patterns of influenza type-A virus (IAV) strains that previously circulated among African human populations between 1993 and 2018. We found that sequencing and genomic characterization of Africa IAVs have increased over the years. However, the implementation of viral whole-genome sequencing using next-generation sequencing (NGS) technology remains low in Africa.

Sequence comparisons showed that IAV strains sampled in Africa evolved continuously, acquiring unique amino acid substitutions in their surface, transmembrane, and internal proteins relative to vaccine strains recommended for use every year. The substitutions affected all the five major antigenic sites (Sa, Sb, Ca_1_, Ca_2_, and Cb) and (A, B, C, D, and E) of the H1 and H3 proteins, respectively. Seasonal H1N1 viruses went extinct in 2009 as a result of host antibodies elicited by the novel pH1N1pdm09 viruses [109]. Both Africa [52,54,55,66] and global pH1N1pdm09 viruses [110,111,112] underwent rapid diversification driven by stochastic substitutions and increased transmission reflected as comb-like phylogenies. Seasonal H1N1pdm09 formed ladder-like phylogenies resulting from immunological pressures due to natural infection or vaccination [98,112,113,114]. Substitutions at antigenic site B (positions 155, 156, 158, 159, 189, and 193) responsible for the antigenic drift and fitness among global 1968–2003 season H3N2 viruses [12,115,116,117,118] were also observed among Africa H3N2 strains [70,73,80,88,89,97,101,102,105]. Both the antigenic drift and annual variation in dominant antigenic sites among Africa IAVs resulted in sub-optimal vaccine protectiveness against circulating strains [78,79,80] hence the need to update vaccine strain compositions every year.

IAVs also evolve from vaccines by altering potential N-glycosylation sites (pNGSs). For example, the earliest Hong Kong H3N2 strain, A/HK/1968(H3N2), had two pNGSs (81 and 165), while strains sampled later gained 10–11 and 8 pNGSs in their H3 and N2 proteins, respectively [15,16,119]. We observed several substitutions resulting in gain (S45N, S124N, A128T, K144N, F159Y, K160T, N225D) and loss (L3I, N144S) of pNGS among Africa H3N2 strains [67,78,79,80,81,93,95,96,97,101], similar to global strains [15]. The HA receptor-binding sites among African IAV strains mostly remained conserved [73,74,81]. Africa pH1N1pdm09 strains had one mutated N1 antigenic site (N248D) [47,48,54,55,56,104] also observed in the USA and reported to alter the antibody recognition site hence affecting the A/California/07/2009(H1N1) vaccine efficacy [120].

Despite the availability of antiviral drugs in 65% (19/31) of African countries as of January 2013, drug use remains very low [121]; hence, drug-resistant variants may have been largely imported into Africa. There was a drastic increase in the global frequency of adamantane-resistant H1N1 (0.3% to 15.6%) and H3N2 (15% to 90.6%) viruses during the 2000–2006 and 2004–2006 seasons, respectively [122]. However, all Africa H1N1 and H3N2 strains sampled during the 2004–2009 and 2004–2006 season, respectively, remained sensitive to adamantanes [41,87,105]. Adamantane-resistant H3N2 strains were introduced in Africa in 2008 [72,73,78,79,105]. The majority of H1N1pm09 and H3N2 strains sampled in Africa and globally between 2009 and 2019 were adamantane-resistant [110,123,124,125]. A new population of adamantane-resistant variants with mutated M2 positions 26, 27, 30, 34, and 38 emerged and circulated in Africa and globally from 2011 through 2016 [122,126,127].

Neuraminidase inhibitors (NAIs), oseltamivir, and zanamivir started to be used between 1999 and 2002 [122]. The first oseltamivir-resistant H1N1-H274Y variants were sampled in France in 2006 (A/Lyon/381/2006(H1N1) [128] and Australia between October 2007 and April 2008 before their introduction in South Africa in 2008 [39,104]. A total of 90% of global H1N1 strains sampled between 2008 and 2009 were oseltamivir-resistant but sensitive to zanamivir [122]. Oseltamivir-resistant H1N1pdm09 strains were sporadically circulated in South Africa [43], globally between 2009 and 2011 [64,95,122], the Middle East and North Africa in 2015 [91], and Myanmar in 2017 [129]. However, there is no sufficient evidence that the resistant H1N1pdm09 variants resulted from oseltamivir treatment. Our analysis shows that the majority of African and global H1N1pdm09 and H3N2 strains sampled during the 2009–2018 season remained sensitive to NAI drugs, making oseltamivir suitable for pre- or post-exposure prophylaxis treatment.

Time-resolved phylogenies showed the cocirculation of multiple IAV lineages and clades each year in Africa as observed globally [130,131]. Africa IAVs phylogenetically mixed with earlier global strains, suggestive of viral introductions into Africa from the rest of the world. Circulating strains in a given year replaced strains in the previous year. However, some H1N1pdm09 lineages persisted through consecutive seasons in the United Kingdom [132], China [113], and Western Africa [60]. Phylogeographical and tree trunk analysis confirmed significant viral migration from South Asia and Europe to Africa (Bayes Factor > 1000) but low migration rates from Africa to elsewhere [21,98,112,133]. Another study by Chan et al. fitted a probabilistic model on the global H1N1 and H3N2 virus HA1 sequences and confirmed Europe and North America as the main importers of IAVs to Africa [86]. Together, the studies confirm Africa as a sink for new IAVs. However, this pattern could be an artifact of inconsistent and insufficient viral sampling in Africa [134].

Molecular dating estimated the ages of TMRCA for Africa and global pH1N1pdm09 strains earlier than their first laboratory detection [55,56,112,135,136]. Following their introduction, IAVs continued to evolve and spread within and between African households, countries, and regions, showing a temporal and geographical evolution structure [47,53,55,60,62,102,103] consistent with previous studies [137]. Global H3N2 and H1N1pdm09 viruses sampled during the 2000–2012 and 2009–2019 season, respectively, had their H3, H1, and N1 genes evolve at mean rate ranges of 4.8–5.2 × 10^−3^, 4.4–5.34 × 10^−3^, and 3.8–5.21 × 10^−3^, respectively [112,113,124,133]. Kenya pH1N1pdm09 virus H1 and N1 and 2007–2013 season H3 genes evolved at comparable mean rates at 5.58 × 10^−3^ (2.75–9.28) and 4.07 × 10^−3^ (1.47–7.73) and 4.17 × 10^−3^ (3.09–5.31) subs/site/year, respectively [54,55,80], confirming a relatively constant viral evolution rate [138]. The surface genes of IAVs evolve more than internal genes because they are subjected to strong immunological selection pressures. All Africa and global 2007–2013 season H3N2 [80] and 2009–2019 season H1N1pdm09 strains [61,66,112,113,139] had their HA genes under negative selection with dN/dS < 1. Except Venter et al. reported dN/dS of 1.6 among HA1 genes of global H1N1pdm09 strains sampled in 2011 [66]. Skyride analysis showed strong oscillations of population sizes of both H1N1pdm09 and H3N2 strains sampled in Africa [54] and globally [113,140,141,142] between 2007 and 2019, indicating that viruses underwent bottlenecks during their evolution and transmission.

Despite the similarity in circulating viral clades, evolutionary rates, and population dynamics, we observed some differences in viral evolution and transmission patterns between Africa and elsewhere. First, the seasonal H1N1 strains went extinct in Africa upon the emergence of pH1N1pdm09 in 2009 but were sporadically sampled elsewhere as of May 2010 [3]. Second, our results show only H1N1 strains belonging to clades 1, 2B, and 2C circulated in Africa between 2001 and 2008. However, additional clades 1, 2A, 2C.1, 2B.1, 2C.2, and 2B.2 circulated in Taiwan between 2005 and 2008 [143]. Third, novel reassortant H1N2 viruses circulated in 2000 in Thailand before they were detected in the USA, Europe, and Africa during the 2001–2003 seasons [89,144,145,146]. Fourth, out of the seven global pandemic H1NIpdm09 genetic clades(1–7) [114], only clades 2, 3, 5, 6, and 7 viruses were pandemic in Africa. Clades 4 and A/Madrid/SO8171/2010-clade strains emerged later in Africa in 2011 [50,51,147]. Fifth, novel H1N1pdm09 clades 8 and 9 circulated only in Africa [50,51,95,97,102,148]. Sixth, H3N2 virus clades 3C.2a and 3C.3a circulated in Switzerland in 2013 before their emergence in Africa in 2014 [77]. Although clades 3C.3a and 3C.2a continued to co-circulate globally through 2018 [15], no 3C.3a-like strain was sampled in Africa after 2016. These differences may be due to the inconsistent and under-sampling of IAVs in Africa. Most studies sampled specific geographical locations, especially urban or cities, and over short periods. In contrast, some countries were un-sampled, creating geographical and temporal sampling biases in our results [134].

Various automated methods based on phylogenetic incongruence, TMRCAs, genetic distances, and coalescent modeling have detected intra- and inter-subtype reassortment among global IAVs [26,113,138,139,149,150,151,152,153]. Global H3N2 strains reassorted more at a rate of 0.3–0.55 events per lineage per year than the H1N1pdm09 (0.1–0.45) [151]. However, reassortment among Africa IAVs has largely been inferred manually based on viral sequences occupying different positions between phylogenies of two or more gene segments [58,60,73,78]. Owuor used the automated reassortment detection tool (GiRaF) and identified no reassortant H3N2 strains in Kilifi (coastal Kenya) between December 2015 and December 2016 [98]. Contrastingly, our ongoing GIRAF analysis of ~1500 Africa virus whole genomes sampled between 1994 and 2020 identified ~573 reassortants, some of which circulated in Kilifi [154]. Such differences could be due to different sampling periods, given that the number of reassortants observed depends on the number of genomes sampled (genomic diversity) [150,154]. Inter-subtype reassortment events observed in Africa between 2009 and 2013 involved human-avian and human-human viral gene exchanges [72,100,105]. However, resulting reassortants could have been less fit to circulate [155]. Africa and global H1N1pdm09 strains sampled in humans and swine mixed phylogenetically indicative of a bidirectional zoonosis and continuous evolution of H1N1pdm09 strains among swine worldwide [42].

## 5. Conclusions

Phylogenetic and phylogeographic analyses showed that multiple introductions of new strains from outside Africa and extensive local viral transmission sustain the high genomic diversity, continuous antigenic drift, and persistence of influenza viruses among African human populations. Circulating strains had several new amino acid substitutions affecting major antigenic and N-linked glycosylation sites in their hemagglutinin (HA) proteins, which could dramatically affect antibody recognition among vaccinated and exposed unvaccinated populations. The role of reassortment and zoonosis in the evolution and diversification of IAVs in Africa needs to be determined. We observed substitutions and clades and persistent viral lineages unique to Africa. Therefore, Africa’s contribution to the global influenza ecology should not be understated. Thus, there is a need to expand influenza surveillance across Africa and prioritize routine whole-genome sequencing using modern NGS technology and genomic analysis to monitor circulating strains and early detection of emerging ones. Such knowledge could inform public health policies and appropriate vaccine development and selection.

## Figures and Tables

**Figure 1 microorganisms-10-00900-f001:**
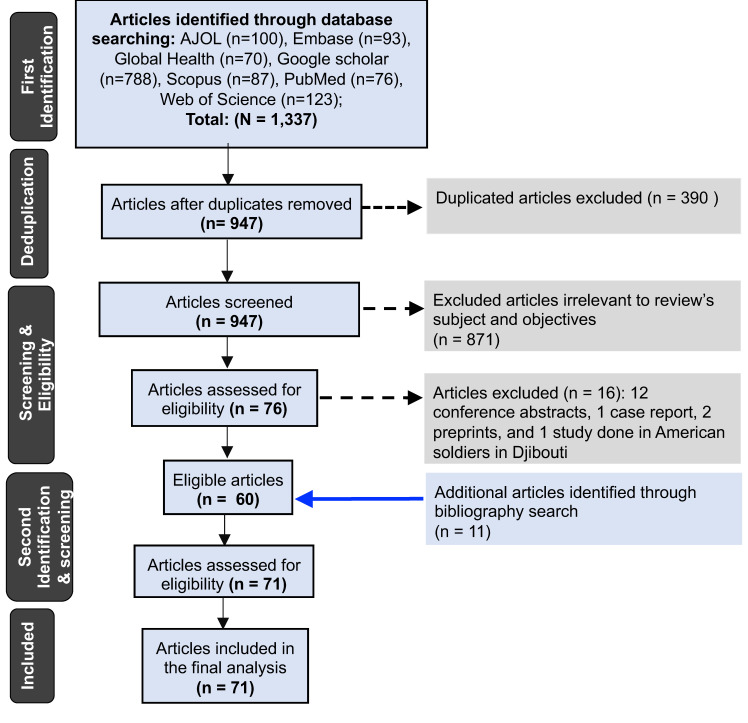
Flowchart showing the exclusion and inclusion criteria of studies analyzed.

**Figure 2 microorganisms-10-00900-f002:**
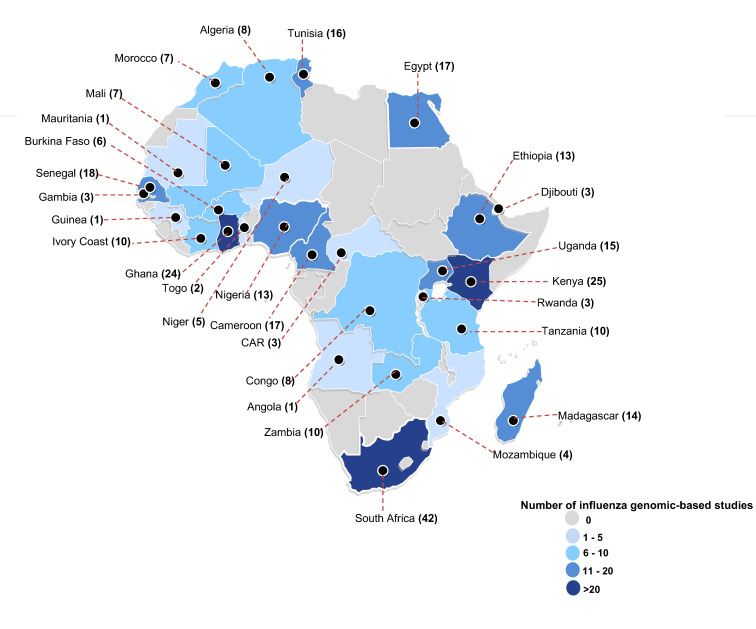
The geographical distribution of studies that did genomic characterization of influenza type-A viruses (H1N1, H1N1pdm09, and H3N2) sampled in Africa. Each country is highlighted based on the absolute number of studies that analyzed sequences of influenza viruses collected from that country. For each country, the study count also includes any study that included at least one sequence from that country in their virus clade classification using the European Center for Disease Control (ECDC) guidelines [27]. Abbreviations: CAR = Central African Republic. Countries not shown: Cape Verde (*n* = 1), Reunion (*n* = 3), Seychelles (*n* = 4), Mauritania (*n* = 1), Mauritius (*n* = 10), and Mayotte (*n* = 1).

**Figure 3 microorganisms-10-00900-f003:**
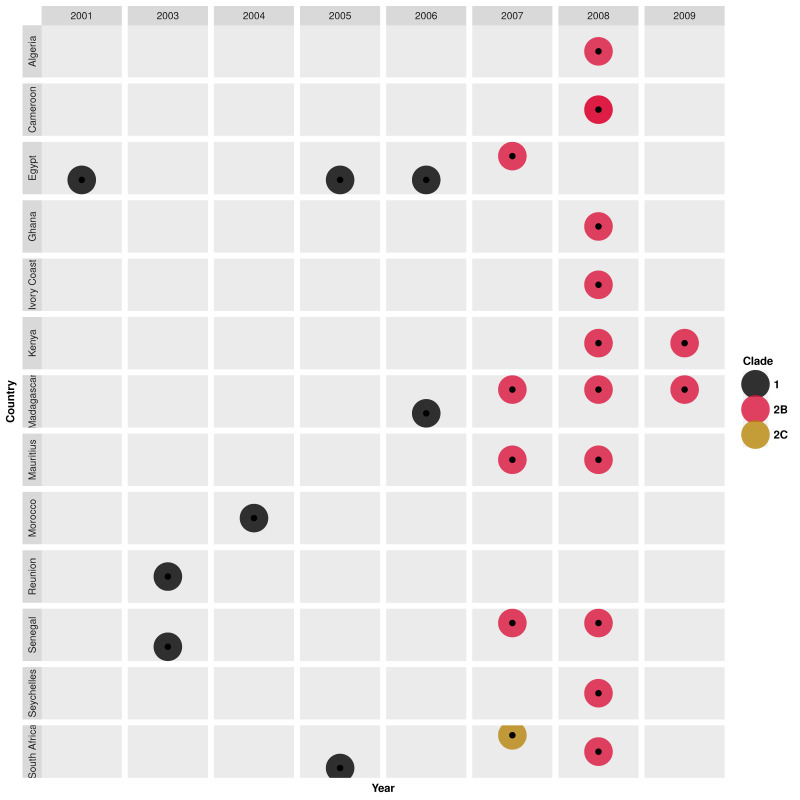
Temporal and geographical distribution of genetic clades of seasonal H1N1 virus strains that circulated in Africa between 2001 and 2009. Details on the characteristic genomic markers (amino acid substitutions in the HA1 proteins) for each clade are described in Appendix A.

**Figure 4 microorganisms-10-00900-f004:**
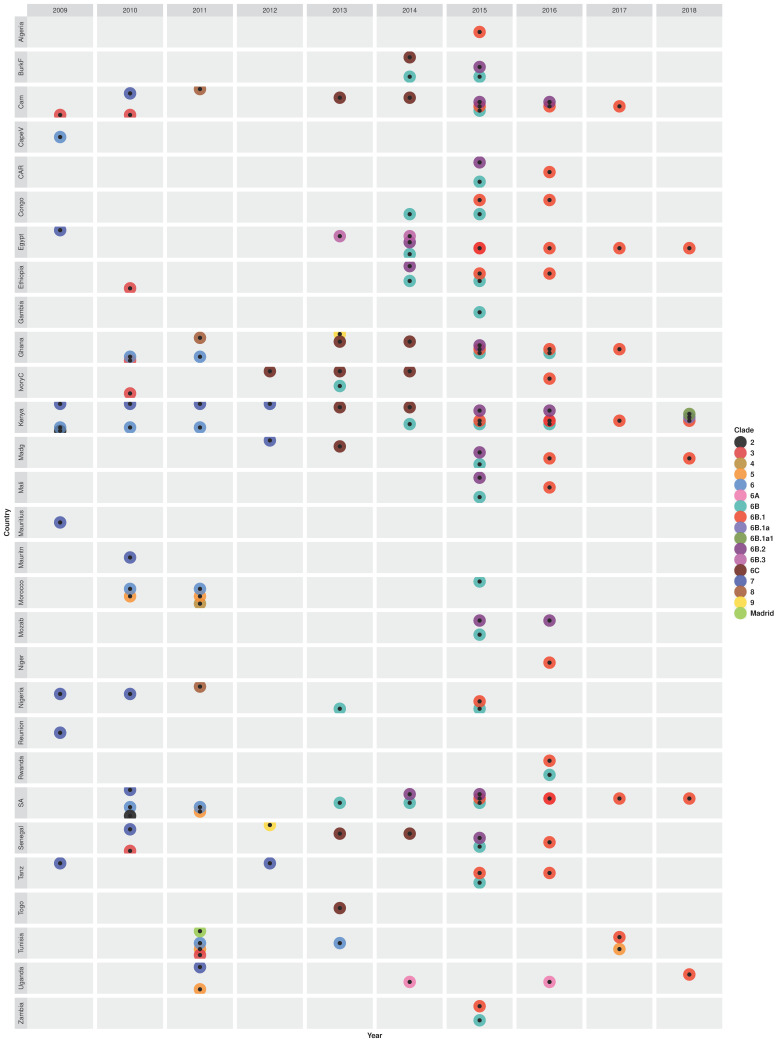
Temporal and geographical distribution of genetic clades of pandemic H1N1pdm09 (2009–2010) and seasonal H1N1pdm09 (2011 onwards) virus strains that circulated in Africa between 2009 and 2018. Abbreviations: Burkina Faso (BurkF), Cameroon (Cam), Cape Verde (CapeV), Central African Republic (CAR), Ivory Coast (IvoryC), Madagascar (Madg), Mauritania (Mauritn), Mozambique (Mozab), South Africa (SA), Tanzania (Tanz), and A/Madrid/SO8171/2010(H1N1)-like clade (Madrid). Details on the characteristic genomic markers (amino acid substitutions in the HA1 proteins) for each clade are described in Appendix A.

**Figure 5 microorganisms-10-00900-f005:**
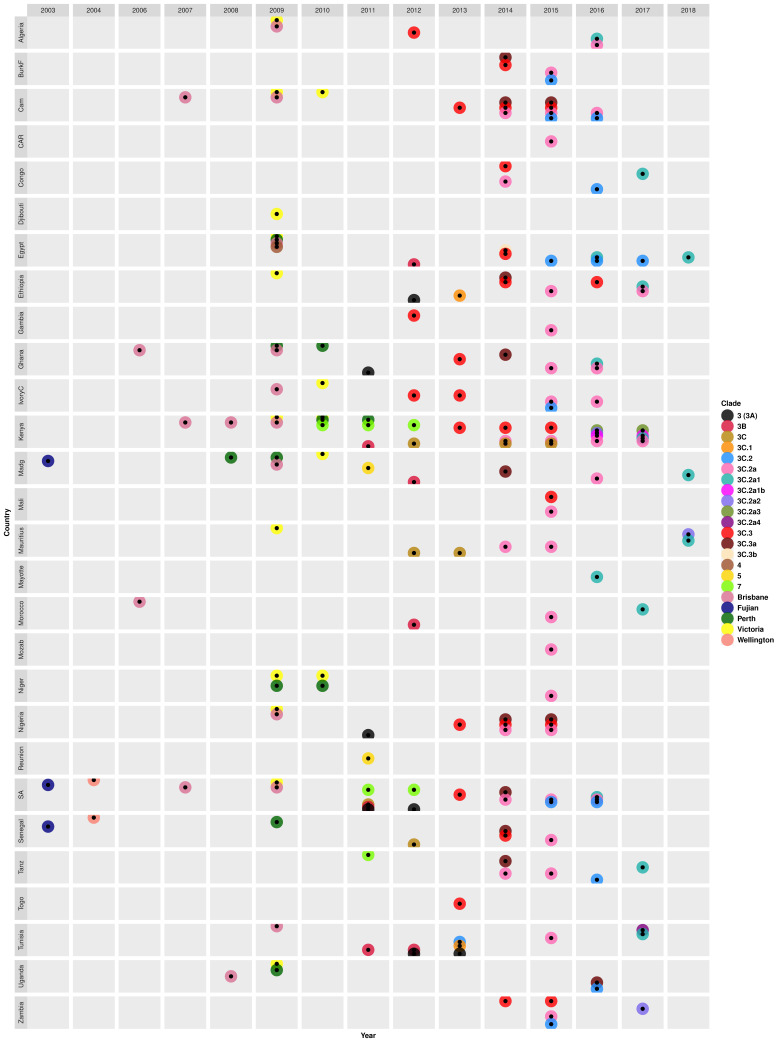
Temporal and geographical distribution of genetic clades of seasonal H3N2 virus strains that circulated in Africa between 2003 and 2018. Abbreviations: Burkina Faso (BurkF), Cameroon (Cam), Cape Verde (CapeV), Central African Republic (CAR), Ivory Coast (IvoryC), Madagascar (Madg), Mozambique (Mozab), South Africa (SA), Tanzania (Tanz). A/Brisbane/10/2007(H3N2)-like clade (Brisbane), A/Fujian/411/2002(H3N2)-like clade (Fujian), A/Perth/16/2009(H3N2)-like clade (Perth), A/Victoria/208/2009(H3N2-like clade (Victoria), and A/Wellington/1/2004(H3N2)-like clade (Wellington). Details on the characteristic genomic markers (amino acid substitutions in the HA1 proteins) for each clade are described in Appendix A.

**Table 1 microorganisms-10-00900-t001:** Characteristics of eligible studies analyzed.

Study (Reference)	Study Type	Location	Sampling Dates	Population; Setting; Viral Detection Method	Sample Selection Criteria for Sequencing or Analysis	No. of Samples Sequenced; Sequencing Technology and Location	Gene/s Sequenced and Analyzed	Analysis Type	Patient’s Variables Collected
**H1N1 viruses**									
Besselaar (2008) [37]	Observational study nested in an active national sentinel surveillance program	South Africa	May–July 2008	ARI patients of all ages; sentinel site surveillance program; shell vial assay or cell culture	All isolated viruses were sequenced	45; sequencing technology and location not described	HA, NA	Mutations; drug sensitivity; genomic variation	3, 4
Bulimo (2012) [38]	Observational study nested in a sentinel surveillance program	Kenya	January 2007–November 2008	ILI outpatients aged >2 months; sentinel site surveillance system; RT-PCR	Randomly selected virus isolates based on the month and location of sampling and sequenced all those that were positive for H1N1; included online HA1 sequences from South Africa, Malaysia, and Thailand	42; Sanger; Texas, USA (abroad)	HA1	Mutations; genomic variation; phylogenetics (Bayesian)	3, 4
Hurt (2009) [39]	Observational study nested in the WHO global influenza surveillance network	Global; Australia, South East Asia, Oceania, and South Africa	October 2007–November 2008	Populations not described; viruses sampled through influenza surveillance networks in 10 countries, subtyped, and submitted to the WHO Influenza Centre, Melbourne; viral detection method not described	Selected viruses with reduced susceptibility to oseltamivir (high IC_50_ values)	264; inclusive of 26 viruses from South Africa; pyrosequencing; Melbourne (abroad)	HA, NA	Mutations: drug sensitivity; phylogenetics (ML)	3, 4
Njouom (2010) [40]	Observational study nested in the national influenza surveillance network	Yaounde, Cameroon	November 2007–October 2008	ILI outpatients of all ages; national influenza surveillance network sampled 7 sentinel sites across Younde (Cameroon); RT-PCR and HAI on cell culture	All viruses successfully isolated in cell cultures were sequenced	10; technology not described; London, U.K. (abroad)	HA, NA, MP	Mutations: drug sensitivity; antigenic inhibition assay; phylogenetics	3, 4
Dia (2013) [41]	Observational study nested in an influenza surveillance network	Senegal	July–September 2008	ILI outpatients aged 2 months to 48 years; sentinel surveillance system; HAI on cell culture	Randomly selected; included 27 viruses isolated earlier in 2007	15; Sanger; London, U.K. (abroad)	HA1, NA, M2	Mutations; drug sensitivity; antigenic analysis; phylogenetics (ML)	3, 4
**H1N1pdm09 viruses**									
Adeola (2019) [42]	Independent syndromic survey	Nigeria and Ghana	January 2014–March 2015 (swine) and December 2015–March 2016 (human)	Swine handlers with ILI and asymptomatic swine in pig farms and Abattoirs in Ibadan (Nigeria) and Kumasi (Ghana); RT-PCR	Sequenced all (3) H1N1pdm09-positive samples observed among the 32 swine handlers that consented to test for influenza	3 Human (Ghana (2), Nigeria (1)) and 3 Swine (Ghana); Sanger; Nigeria (local); included Africa and global human (2009–2014) and swine (2011–2018) viral MP sequences from GenBank	MP	Phylogenetics (ML); zoonosis	1, 2, 3, 4
Aspinall (2013)[43]	Independent longitudinal observational trial	South Africa	2009–2010	Patients with fever and positive influenza rapid antigen tests of one year and older (1–64 years); community-based trial; RT-PCR	All H1N1pdm09 positives were sequenced	44; technology not described; South Africa (local)	NA	Mutations: drug sensitivity; neuraminidase inhibition assay (NAI)	1, 2, 3, 4, 5, 6, 8, 10, 11, 13
Ayim-Akonor (2020) [44]	Independent active cross-sectional study	Ghana	April–July 2016 and December 2016–February 2017	Asymptomatic farmers and apparently healthy swine aged 6–24 weeks at piggery farms in Ashanti region of Ghana; RT-PCR	Sequenced all H1N1pdm09 influenza-positive swine samples with PCR CT value <25	8; Sanger; Germany (abroad); included Africa and global human and swine viral sequences from GISAID sampled in 2016–2017	3 WGs and 11 PGs (HA and NA)	Antigenic analysis; phylogenetics (ML); zoonosis	1, 2, 3, 4
Ben Hamed (2021) [45]	Independent observational study	Tunisia	2017–2018	SARI patients aged (3 months-80 years) in the Sahel region (Monastir) of Tunisia; RT-PCR	Sequenced viral samples for patients with fatal and severe cases with ≤5 days of infection and the best PCR bands	7; Sanger; Tunisia (local); included global N1 sequences sampled in 2010–2011	NA	Phylogenetics (ML)	1, 2, 3, 4, 6, 8, 13
Bonney (2012) [46]	Observational study nested in the national influenza surveillance system	Ghana	January 2008–December 2010	0–10-years-old children with ILI; influenza surveillance system; RT-PCR	Selection criteria not described	13; technology not described; London, U.K. (abroad)	NA	Drug sensitivity (sialidase inhibition assay); antigenic analysis (HAI); phylogenetics	1, 2, 3, 4, 8
Byarugaba (2016) [47]	Observational study nested in a sentinel surveillance program	Uganda	July 2009–May 2011	ILI outpatients for all ages; hospital-based sentinel surveillance system (Mulago, Jinja, Bugiri, and Gulu); RT-PCR	Not described	19; Sanger; Memphis, TN, USA (abroad)	WGs	Mutations; drug sensitivity; antigenic analysis (HAI); phylogenetics	1, 2, 3, 4
Dia (2013) [48]	Prospective observational study nested in a surveillance program in Senegal	West Africa; (Cape Verde, Guinea, Mauritania, and Senegal)	June 2009–October 2010	ILI patients of all ages; surveillance program in Senegal and for other countries samples were collected in the context of the pandemic mostly from health centers in capital cities; RT-PCR	Selected samples based on time and country, and with PCR Ct ≤ 30	14; Senegal (7), Mauritania (6), Cape Verde (1) Sanger; London, U.K. (abroad)	HA, NA	Mutations; drug sensitivity; phylogenetics (ML)	1, 2, 3, 4, 8
El Moussi (2013) [49]	Observational study nested in a viroloigcal surveillance network	Tunisia	2009–2011	ILI and SARI patients of all ages; Virological surveillance network; RT-PCR	Selected samples based on clinical symptoms (42 severe and fatal and 8 mild)	50; targeted hemagglutinin analysis and Sanger; Tunisia (local)	HA (931 bps)	Mutations	1, 2, 3, 4, 8
El Moussi (2013) [50]	Observational study nested in a sentinel centre and hospital surveillance	Tunisia	May 2009–December 2011	ILI patients of all ages; sentinel centre and a hospital surveillance; RT-PCR	Selection criteria not described	50; Sanger, Tunisia (local)	HA (931 bps)	Mutations; phylogenetic analysis (ML)	1,3, 4, 8, 10
El Moussi (2013) [51]	Observational study nested in an influenza surveillance program	Tunisia	2008–2011	ILI and SARI patients of all ages; surveillance system; RT-PCR	Randomly selected regardless of the clinical symptoms	50; technology not described; London, U.K. (abroad)	HA (931 bps), NA (836 bps)	Mutations; drug sensitivity; phylogenetics	1, 2, 3, 4, 8
El Rhaffouli (2013) [52]	Independent observational surveillance	Morocco	June 2009–May 2011	ILI patients of all ages; independent surveillance at a military hospital; RT-PCR	Samples selected based on patient’s region and epidemic phase; included 14 HA1 sequences from GenBank sampled from Casablanca	22; Sanger; Morocco (local)	HA1	Mutations; phylogenetics (NJ)	1, 2, 3, 4, 8, 9, 13
Gachara (2011) [53]	Independent observational study	Kenya	July–December 2009	Patients of all ages that met the case definition criteria for pandemic flu developed by the Ministry of Public Health; RT-PCR	Viruses selected based on the Kenya’s provinces; included 10 GenBank sequences (Nigeria (3), Senegal (4), Ethiopia (2), and Mali (1))	31; technology not described; Kenya (local)	NS; (NS1 (219 bps), NS2 (121 bps))	Mutations; phylogenetics (NJ)	3, 4
Gachara (2014) [54]	Cross-sectional retrospective study nested in the national pandemic outbreak surveillance system; PhD thesis	Kenya	July 2009–August 2010	ARI Patients of all ages; global influenza pandemic surveillance system; RT-PCR	Samples selected based on time (week) and geographical region; included 320 global sequences from IRD	40; Sanger; Kenya (local)	WGs; Concatenated all eight genes (concat-8)	Mutations; drug sensitivity; phylogenetics (Bayesian); phylodynamics; population dynamics	1, 2, 3, 4
Gachara (2016) [55]	Observational study nested in the global influenza pandemic response system	Kenya	July 2009–August 2010	ILI patients of all ages; global influenza pandemic surveillance system; RT-PCR	Selected samples based on time and site (not more than 2 isolates per site per week)	40, Sanger, Kenya (local)	WGs (concat-8)	Mutations; drug sensitivity; reassortment; phylogenetics (Bayesian); phylodynamics	3, 4
Meseko (2015) [56]	Independent observational study	Africa; (Tunisia Nigeria, South Africa, Cameroon, Angola, Uganda, Ghana, Algeria, Djibouti, Egypt, Ethiopia, Ivory Coast, Zambia, Mali, and Togo)	2009–2013	Online data	Downloaded Africa H1N1pdm09 viral sequences from GenBank and GISAID	Downloaded 115 HA and 75 NA human viral sequences; included 4 swine H1N1pdm09 viral sequences (2010–2013)	HA, NA	Mutations; phylogenetics (NJ); zoonosis	3, 4
Meseko (2019) [57]	Independent observational study	Nigeria	July 2010–June 2012	Swine with influenza-like signs sampled in a multi-complex commercial piggery in Lagos; RT-PCR	Sequenced swine H1N1pdm09 viruses that were successfully cultured; included Africa and global human, swine, and avian viral sequences from GISAID and GenBank	Sequenced 12 swine viruses; Sanger (local)	WGs (*n* = 3) and PGs (*n* = 12)	Mutations; phylogenetics (ML); zoonosis	3, 4
Monamele (2019) [58]	Observational study nested in an influenza surveillance system	Cameroon	January 2014–June 2016	ILI and SARI patients for all ages; influenza surveillance system; RT-PCR	Randomly selected 23 samples with PCR CT < 30; Included 22 Cameroon sequences from GISAID collected in the same period	17 successfully sequenced (2014 (2), 2015 (30), and 2016 (7)); sequencing technology not described; GENEWIZ, U.K. (abroad)	HA, NA, MP	Mutations; N-glycosylation site prediction; vaccine efficacy using P_epitome_ model; drug sensitivity; phylogenetics (ML)	3, 4
Nakoune (2013) [59]	Independent prospective surveillance study	Central African Republic (CAR)	January–December 2010	ILI and SARI children aged 0–15 years; sentinel site surveillance network; RT-PCR	Sequenced all H1N1pdm09 positives	5; technology not described; Germany (abroad)	HA (253 bps)	Sequence homology	1, 2, 3, 4, 6
Nelson (2014) [60]	Independent observational study	Global; inclusive of 18 African countries (Algeria, Morocco, Egypt, Burkina Faso, Cameroon, Ivory Coast, Ghana, Niger, Nigeria, Senegal, Djibouti, Ethiopia, Kenya, Madagascar, Tanzania, Uganda, Zambia, South Africa)	2009–2013	Online sequences from GISAID; online	Downloaded only full-length HA and NA sequences	299; online sequences; GISAID	HA, NA	Phylogenetics (ML)	3, 4
Opanda (2020) [61]	Observational study nested in a country-wide human respiratory viruses sentinel surveillance network	Kenya	2015–2018	ILI and SARI patients of all ages; hospital-based sentinel surveillance network; RT-PCR	Sequenced all viruses successfully isolated; (2015 (5), 2016 (2), 2017 (2), 2018 (29))	38; Sanger; Kenya (local)	HA1	Mutations; N-glycosylation site and vaccine efficacy prediction; antigenic analysis (HAI); phylogenetics (Bayesian); phylodynamics: natural selection pressure	3, 4
Orelle (2012) [62]	Observational study nested in the national sentinel surveillance network	Madagascar	August 2009–February 2010	ILI patients of all ages; national sentinel site surveillance network; RT-PCR	Selection criteria not described; included online viral sequences from Madagascar (2011, *n* = 5) and the global (2009–2010)	26 HA and 11 NA; technology not described; South Africa (local) and London, U.K. (abroad)	HA1, NA	Mutations; drug sensitivity (sialidase inhibition assay); antigenic analysis (HAI); phylogenetics (NJ)	1, 2, 3, 4, 8
Pascalis (2012) [63]	Independent prospective community household-based cohort study	Reunion	July–October 2009	ILI patients of all ages sampled from 772 households of 2,164 healthy individuals across the island; cohort study; RT-PCR	Selected samples to reflect epidemiological and temporal dynamics of the pandemic (during and post) in the cohort; included global WGs from GenBank and GISAID	28; technology not described; Reunion (local)	15 WGs (concat-8); 13 PGs (concat-6: PA, HA, NP, NA, M and NS)	Mutations; drug sensitivity; phylogenetic analysis (ML and Bayesian); phylodynamics; molecular dating	1, 2, 3, 4, 8, 12
Quiliano (2013) [64]	Independent observational study	Global; (America, Asia, Oceania, Europa, Africa, and Mexico)	2009–April 2011	Online sequences from Influenza virus sequence database; online	Complete gene or protein sequences	3740 (Africa (59), America (2298), Asia (521), Oceania (89), Europe (772)) complete NA genes	NA	Mutations; drug sensitivity; statistical analysis	3, 4
Valley-Omar (2015) [65]	Independent hospital-based observational study	Cape Town, South Africa	1 April–31 July 2011	Frozen influenza-positive swabs sampled by the national health laboratory from children admitted in 4 Cape Town hospitals; PCR	Sequenced all viral samples collected in the 4-month; included 105 South Africa and globe sequences sampled in 2011 from GenBank	18; Sanger; South Africa (local)	HA (379 to 1204 nts)	Phylogenetics (ML and Bayesian); phylodynamics	1, 2,3,4, 8
Venter (2012) [66]	Observational study nested in multiple (3) influenza surveillance systems	South Africa	July 2009–December 2010	ILI and SARI patients of all ages; multiple (3) influenza surveillance systems; RT-PCR	Sequenced viruses selected based on the geographical location and year of sampling	72 HA, 118 NA, 30 PB2; Sanger; South Africa (local)	PB2, HA, NA	Mutations; drug sensitivity; antigenic analysis (HAI); phylogenetics (NJ and ML); phylodynamics; natural selection pressure; molecular dating; evolutionary rates	3, 4, 8
**H3N2 viruses**									
Aboualy (2018) [67]	Observational study nested in a national influenza surveillance program	Egypt	October–December 2014	ILI outpatients of all ages; national influenza surveillance program; RT-PCR	Sequenced all viruses that were non-agglutinating in the cell cultures	4; Sanger; Egypt (local)	HA, NA	Mutations	3, 4
Besselaar (1996) [68]	Observational study nested in a viral watch program	South Africa	June 1993–September 1994	Archived viral samples from ARI patients of all ages (white and black); Witwatersrand Viral Watch Program; HAI on cell culture	Selected viruses based on the time of collection: beginning, middle, and end of each influenza season	9; Sanger; South Africa (local)	HA1 (500 bps) spanning the receptor binding and antigenic sites	Mutations	3, 4
Besselaar (1999) [69]	Observational study nested in an active surveillance program and routine diagnosis system	South Africa	1997–1998	ARI patients of all ages and infants with SARI; both surveillance and routine diagnosis programs; shell vial method or cell culture	Selected viruses based on the time of collection: beginning, middle, and end of influenza seasons	Sanger; South Africa (local)	HA1	Mutations; antigenic analysis (HAI)	2, 3, 4, 8, 9
Besselaar (2004) [70]	Observational study nested in an active surveillance program	Pretoria and surrounding areas, South Africa	25 May–7 June 2003	Patients of all ages with Acute febrile illnesses living in a police residential college in Pretoria and active surveillance program in Johannesburg, Middleburg, and Vanderbijlpark; RT-PCR	Selected 20 (Pretoria) and 30 sporadic (surrounding areas) H3N2-positive samples for sequencing; selection criteria not described; included South Africa viral sequences sampled in 2002	50; Sanger; South Africa (local)	HA1 (1,073 bps)	Mutation: antigenic analysis (HAI); phylogenetics (NJ)	3, 4
Bulimo (2008) [71]	Observational study nested in an active multiple-institute influenza surveillance system	Kenya	July 2006–April 2007	ILI outpatients older than 2 months; influenza surveillance system; HAI on cell culture	Sequenced all H3N2 positives	9 (2006 (4), 2007 (5)): Sanger; Texas, USA (abroad)	HA1	Mutations; antigenic analysis (HAI); phylogenetic analysis (NJ)	1, 2, 3, 4, 8, 9, 12, 13, 14
Bulimo (2012) [72]	Observational study nested in a hospital surveillance system	Kenya	October–December 2010	ILI patients of > 2 months of ages; hospital surveillance system; RT-PCR	Sequenced all the H3N2 positives confirmed by RT-PCR	32; Sanger; Kenya (local)	HA, NA, MP	Antigenic analysis (HAI); phylogenetics (Bayesian); reassortment	3, 4
Byarugaba(2011) [73]	Observational study nested in a routine hospital-based influenza surveillance system	Mulago and Kayunga, Uganda	1 October 2008–30 September 2009	ILI outpatients aged 6 months and above; hospital-based influenza surveillance system; RT-PCR	Sequenced all the H3N2 positives confirmed by RT-PCR	59 (Mulago (54) and Kayunga (5)); NGS Illumina and Sanger; Memphis, Tennessee, USA (abroad)	WGs	Mutations; homology; drug sensitivity; phylogenetics (ML)	3, 4
El Moussi (2014) [74]	Independent observational study	Tunisia	29 January–February 2013	H3N2-positive patients of all ages with mild, severe, and fatal cases; sampling setting not described; RT-PCR	Selection criteria not described	5; Sanger; Tunisia (local)	HA	N-glycosylation site prediction	1, 2, 3, 4, 8
Kaira (2011) [75]	Cross-sectional Observational study nested in a routine hospital-based influenza surveillance system; MSc thesis	Mulago and Kayunga, Uganda	October–December 2008	ILI outpatients aged 6 months and above; routine hospital-based influenza surveillance system; RT-PCR	Sequenced all the H3N2 positives confirmed by RT-PCR	50 (Mulago (45) and Kayunga (6)); NGS Illumina; Memphis, Tennessee, USA (abroad)	WGs	Mutations; population genetics analysis using Arlequin software; phylogenetics (ML)	1, 2, 3, 4
Kleynhans (2019) [76]	Independent retrospective cohort combined with a questionnaire-based cross-sectional study	Eastern Cape province (ECP), South Africa	13–29 July 2016	Students with ILI in grade 8–12 and bridge year; included 42 South Africa viruses from GISAID collected on 19 May–8 August 2016; RT-PCR	Selected 19 out of the 27 H3N2 positives for sequencing; no selection criteria described	19; NGS Illumina; South Africa (local)	HA	Phylogenetics (ML)	1, 2, 3, 4, 9, 13
Lemey (2014) [21]	Independent observational study	Global; (Europe, Asia, Oceania USA and Africa (Algeria, Egypt, Madagascar, South Africa, and Saudi Arabia))	2002–2007	Online sequences; online	Downloaded available sequences based on location and year of collection	1529 (Africa (31), Rest of the world (1498))	HA	Generalized linear model (GLM); phylogenetics (Bayesian); phylogeography	3, 4
McAnerney (2015) [77]	Case-control study nested in a sentinel surveillance program	South Africa	May–September 2014	ILI outpatients of all ages with and without PCR-confirmed influenza; sentinel surveillance program; RT-PCR	Selected samples based on phase of the season (beginning, mid, and end)	34 (vaccinated (10) and unvaccinated (24)); technology not described; South Africa (local)	HA1 (850 bps)	Mutations; homology; N-glycosylation sites prediction; antigenic analysis (HAI); phylogenetics (ML)	1, 2, 3, 4, 8, 13
Monamele (2017) [78]	Observational study nested in an influenza surveillance system	Cameroon (Southern regions)	January 2014–June 2016	ILI outpatients of all ages; influenza surveillance system; RT-PCR	Randomly selected H3N2 positives clinical samples with PCR CT < 30 based on their geographical origin and distribution over the sampling period	35 (2014 (6) 2015 (17) and 2016 (12)); sequenced technology not described; GENEWIZ, United Kingdom (abroad)	HA, NA, MP	Mutations; drug sensitivity; N-glycosylation site predication using NetNGlyc 1.0; vaccine efficacy prediction using P-epitope model; phylogenetics (NJ)	3, 4
Njifon (2019) [79]	Observational study nested in an influenza surveillance system	Cameroon (Northern region)	January 2014–June 2016	ILI outpatients of all ages; influenza surveillance system; RT-PCR	Randomly selected H3N2 positives clinical samples with PCR CT < 30 based on their geographical origin and distribution over the sampling period; included 35 sequences from Southern Cameroon sampled in 2014–2016	16; Sanger; GENEWIZ U.K., United Kingdom (abroad)	HA, NA, MP	Mutations; drug sensitivity; vaccine efficacy prediction using P-epitope model; phylogenetics analysis (NJ)	3, 4
Nyang’au (2020) [80]	Independent observational study	Kenya	2007–2013	Online sequences; online	Kenya HA1 sequences from GenBank and GISAID sampled by the National Influenza Centre (NIC); included 56 global viral sequences	115; online sequences	HA (HA1)	Mutations; N-glycosylation site predication; vaccine efficacy predication using the P-epitope model; phylogenetics (Bayesian); phylodynamics: molecular dating, natural selection analysis; evolutionary rates	3, 4
Owuor (2020) [81]	Observational study nested in two hospital-based surveillance systems	Kilifi, Kenya	January 2009–March 2017	Inpatients aged below 5 years and outpatients of all ages sampled in Jan 2009–Dec 2015 through a viral pneumonia surveillance and Dec 2015–March 2017 through an influenza surveillance system, respectively; RT-PCR	Selected 186 of the 292 influenza positives with sufficient RNA for sequencing	142 (H3N2 (101), H1N1pdm09 (41)); NGS (Illumina MiSeq); Kilifi Kenya (local)	142 WGs sequenced but analyzed only complete HA genes	Mutations; N-linked glycosylation site prediction; phylogenetics analysis (ML)	1, 2, 3, 4, 8
Westgeest (2014) [26]	Observational study nested in the global WHO influenza surveillance network	Global (Asia, Europe, America, and South Africa)	1968–2011	Viral samples collected through the WHO Collaborating national influenza surveillance centers; viral detection method not described	Sequenced viruses successfully cultured in chicken eggs by a previous study; included online global sequences inclusive of 3 viruses from South Africa (1 WG and 2 PGs)	284; Sanger and NGS; United States (abroad)	WGs and PGs	Phylogenetics; phylodynamics; molecular dating; reassortment	3, 4
WHO (2003) [82]	Retrospective observational study nested in the WHO surveillance system	Bosobolo, CONGO	November–December 2002	ARI patients of all ages; influenza surveillance system and retrospective morbidity survey; RT-PCR and ELISA immunocapture assay (for swabs), and HAI on blood samples	Selected 6 of the 792 ARI samples for analysis; selection criteria not described; 4 of the 6 were H3N2-positive and were sequenced	4; technology and location not described	HA (HA1), NA	Antigenic analysis (HAI); phylogenetics	3, 4, 8
**H1N1 and H3N2 viruses**									
Barakat (2011) [83]	Observational study nested in the national sentinel influenza surveillance system	Morocco	1996–1998	ILI patients of all ages; national sentinel influenza surveillance system; HAI on cell culture	Viral samples selected based on time and location of sampling	29 (21 H3N2, 5 H1N1, and 3 B); Sanger; Morocco (local)	HA1 (980 bps)	Mutations; antigenic analysis (HAI and NAI); phylogenetics (NJ)	1, 2, 3, 4, 6, 8
Barr (2010) [84]	Retrospective observational study nested in the WHO surveillance system	Global; included Kenya, South Africa	September 2008–February 2009	Viral isolates collected through the WHO Collaborating national influenza surveillance centers;	Not described	Number of sequences not described but included 3 Africa strains (1 H1N1, 2 H3N2, and 1 B); technology and location not described	HA, NA, MP, and for some WG	Mutations; drug sensitivity; antigenic analysis (HAI); phylogenetics (ML)	3, 4
Besselaar (2001) [85]	Observational study nested in an active surveillance program and routine diagnosis systems	South Africa	1997–1999	ILI and SARI patients of all ages; surveillance and routine diagnosis programs; shell vial method or cell culture	Selected based on phase (start, middle, and end) of each year’s influenza season	26 (7 H1N1 and 19) H3N2; Sanger; South Africa (local)	HA1	Phylogenetic analysis (NJ)	3, 4
Chan (2010) [86]	Independent observational and modeling study	Global; (Asia, Europe, America, Oceania, and Africa (Kenya))	2004–2009	Online sequences; all H1N1 and H3N2 viral sequences available on NCBI	All H1N1 and H3N2 viral sequences available on NCBI	>6000 online HA, NA, and HA1; inclusive of ~100 Kenya H3N2 and H1N1 viral HA1 sequences	HA, NA	Probabilistic modeling; network analysis	3, 4
Deyde (2007) [87]	Observational study nested in the WHO global influenza surveillance network	Global; Asia, Europe, North and South America, Oceania, and Africa (Egypt and South Africa)	October 2004–September 2006	Viruses sampled through surveillance programs and laboratories globally and submitted to the WHO Influenza Center at the CDC (Atlanta); HAI on cell culture and RT-PCR	All submitted viruses were screened for adamantane resistance using pyrosequencing; selection criteria for viruses sequenced not described	57 M2 and 72 HA1 genes (H1N1 and H3N2); pyrosequencing and Sanger; Atlanta, USA (abroad)	(HA) HA1 and MP (M2) (44 bps)	Mutations; drug sensitivity; phylogenetics (ML)	3, 4
Heraud (2012) [88]	Observational study nested in a national sentinel surveillance system	Africa; (Cameroon, Ivory Coast, Madagascar, Niger, Seychelles, and Senegal)	2008–2010	ILI outpatients of all ages; national sentinel surveillance systems; RT-PCR and cell culture (virus isolation)	Viruses sequenced were selected based on country	113 H1N1 and 151 H3N2; Sanger; both in Madagascar (local) and London, U.K. (abroad)	HA (HA1, 886 bps), NA	Mutations: drug sensitivity; antigenic analysis (HAI); phylogenetics (Bayesian)	3, 4
Niang (2012) [89]	Observational study nested in the national influenza surveillance network	Dakar, Senegal	January 1996–December 2009	ILI outpatients of all ages; national Influenza surveillance network; RT-PCR and cell culture	Not described	36 (24 H3N2, 9 H1N1, 3 H1N2); Sanger; London, U.K. (abroad)	HA (H3 (160–971 bps), H1 (52–964 bps)), NA	Antigenic analysis (HAI); phylogenetics	1, 2, 3, 4, 8
**H1N1pdm09 and H3N2 viruses**									
Ait-Aissa (2018) [90]	Retrospective observational study nested in the national sentinel surveillance network	Algeria	2009–2014	ILI patients of all ages; sentinel surveillance network; RT-PCR	Selected viruses with high IC_50_ values compared to normal ranges	11 (3 H1N1pdm09, 6 H3N2, and 2 B); technology not described; London (abroad)	NA	Mutations: drug sensitivity; antigenic analysis (NAI); PCR allelic discrimination and sequence analysis to detect H275Y	1, 2, 3, 4, 8
Al Khatib (2019) [91]	Independent observational study	Global; Middle East and North Africa (Tunisia, Egypt)	2009–2017	Online sequences; online	Downloaded all available sequences from MENA region through the Influenza Research Database (IRD); no data were available from Saudi Arabia, Yemen, or Libya	1226 online sequences (512 H1, 239 H3, 343 N1, and 132 N2) were analyzed	HA, NA	Mutations: phylogenetics (ML); phylodynamics: evolutionary rates; selection pressure analysis	3, 4
Barr (2014) [92]	Retrospective observational study nested in the WHO surveillance system	Global; included Egypt, Kenya, Tanzania, Tunisia, Mauritius, Senegal, South Africa	September 2012–February 2013	Clinical specimens or virus isolates collected through the WHO Collaborating national influenza surveillance centers	Not described	379 H1N1pdm09 and 872 H3N2 included 226 Africa strains (35 H1N1pdm09, 115 H3N2, and 76 B); technology and location not described	HA, NA, MP, and for some WG	Mutations; drug sensitivity; antigenic analysis (HAI); phylogenetics (ML)	3, 4
Bulimo (2012) [93]	Observational study nested in the national influenza surveillance network	Kenya	2010–2011	ARI Patients of 2 months and above; national influenza surveillance network; RT-PCR	All influenza positives were analyzed genetically	62 (27 H1N1pdm09, 19 H3N2 and 16 B); Sanger; Kenya (local)	HA1	Mutations; phylogenetics (Bayesian)	3, 4
Kavunga-Membo (2018) [94]	Observational study nested in an influenza sentinel surveillance system	CONGO	January–December 2015	ILI and SARI patients of all ages; influenza sentinel surveillance; RT-PCR	Selection criteria not described	Number of samples shipped for sequencing not reported; technology not clear (Sanger or NGS?); Atlanta, USA (abroad)	HA	Antigenic analysis (NAI); phylogenetics (ML and NJ)	2, 3, 4, 8
Klimov (2012) [95]	Observational study nested in the WHO global influenza surveillance network	Global; inclusive of Africa (Algeria, Cameroon, Ivory Coast, Egypt, Ethiopia, Ghana, Kenya, Madagascar, Niger, Nigeria, Senegal, South Africa, Tanzania, Uganda, Morocco, Tunisia), Asia, America, Europe, and Oceania	February–September 2011	Online sequences; viruses were sequenced from clinical samples and viral isolates from patients of all ages sampled through the global WHO national influenza centers and laboratories within and outside of GISRS	Antigenic analysis (HAI) performed on all 4400 viruses, but selection criteria for viral WGs and PGs chosen for genomic analysis not described	Downloaded > 1600 global H1N1pdm09 and >1000 global H3N2 viral sequences inclusive of 114 H1N1pdm09 and 112 H3N2 from Africa	WGs and PGs (HA or HA1 and NA)	Mutations; drug sensitivity; antigenic analysis (HAI); phylogenetics (ML); vaccine efficacy; phylodynamics; reassortment	3, 4
Mackenzie (2019) [96]	Observational study nested in a population-based surveillance	Gambia	10 February–31 December 2015	In- and out patients with ALRI and suspected pneumonia aged 2–23 months; surveillance system; RT-PCR	Samples selected based on viral load (PCR Ct <30) and distribution across the 11 months of sampling	16 (4 H1N1pdm09, 4 H3N2, and 8 B); Sanger; Gambia (local)	HA	Mutations; N-glycosylation site prediction; phylogenetics	1, 2, 3, 4, 6, 8
Nkwembe (2016) [97]	Observational study nested in the National influenza surveillance system	CONGO	August–December 2014	ILI and SARI patients of all ages; national public health Influenza surveillance system; RT-PCR	Selected samples with PCR cycle threshold <30; included Africa sequences from GISAID	20 (18 H3N2 and 2 H1N1pdm09); NGS (Illumina); Atlanta, USA (abroad)	HA	Antigenic analysis (HAI and NAI); phylogenetics (ML)	3, 4
Owuor (2021) [98]	Observational study nested in five disease surveillance and research programs	Kenya	2009–2018	SARI, pneumonia, ARI, and LRTI patients of all ages sampled through five surveillance and research programs; RT-PCR	Archived viral samples with sufficient volume (≥140 µL) for RNA extraction and sequencing, and positive for IAV	549 (Kilifi (66), Kenya (383), and Africa (100) consisting of H1N1pdm09 (414) and H3N2 (135)); NGS (Illumina); Kilifi, Kenya (local); included online Africa (H1N1pdm09 (155) and H3N2 (281)) and global viral sequences from GISAID	WGs (concat-8)	Phylogenetics (Bayesian); phylodynamics; molecular dating; population dynamics; phylogeography; phylodynamics; reassortment	1, 2, 3, 4, 8
Sanou (2018) [99]	Observational study nested in the national influenza sentinel surveillance system	Burkina Faso	January 2014–December 2015	ILI and SARI patients under 5 years old; influenza sentinel surveillance system; RT-PCR	Influenza A-positive samples with PCR Ct ≤ 30	43 (14 H1N1pdm09 and 29 H3N2); Sanger; Burkina Faso (local)	HA	Phylogenetics (ML)	1, 2, 3, 4, 8
Soli (2019) [100]	Independent observational study	Tunisia	2009–2013	Online sequences for H1N1pdm09, H3N2, and avian H9N2 viruses from NCBI; online	Sequences collected in the 2009–2013 seasons	102 online sequences	HA, NA for all 3 subtypes; PB2, NP, and M for avian H9N2 viruses	Phylogenetics (Bayesian); phylodynamics; Recombination	3, 4
Soliman (2020) [101]	Independent hospital-based observational study	Egypt	January 2015–December 2016	Sampled 60 ILI children under 5 years old at a pediatric hospital in Egypt; RT-PCR	Sequenced viral samples with high IAV-positive signal; included 116 H1N1pdm09 (2009–2017) and 82 H3N2 (2006–2017) from GISAID sampled in Egypt	Sequenced 10 but recovered 5 (3 H3N2 and 2 H1N1pdm09); Sanger; Egypt (local)	HA, NA	Mutations; phylogenetics (ML)	3, 4
Tivane (2018) [102]	Observational study nested in the National Institute of Health (NIH) influenza sentinel surveillance system	Mozambique	January–June 2015	SARI inpatients of ages 0–12 years old; national influenza sentinel surveillance system; RT-PCR	Selected influenza-positive samples with PCR ≤ 30	19 (12 H3N2, 4 H1N1pdm09, 3 B); Sanger; Francis Crick Institute, United Kingdom (abroad)	HA, NA	Mutations; drug sensitivity using fluorescent neuraminidase activity inhibition; antigenic analysis (HAI); phylogenetics (ML)	1, 2, 3, 4, 8
Valley-Omar (2018) [103]	Retrospective observational study nested in a large household-based transmission study	South Africa	May–October 2013	ILI patients of all ages and their contacts irrespective of presence of ILI symptoms; large household transmission study (HTS); RT-PCR	Sequenced viruses from all index and contact samples collected from households with viral transmission	35 (17 H1N1pdm09 and 18 H3N2 from 6 and 8 HHs, respectively); Sanger; South Africa (local)	HA (HA1)	Phylogenetics (ML)	2, 3, 4, 5, 6, 12, 13
**H1N1, H1N1pdm09, and H3N2 viruses**									
Treunicht (2019) [104]	Observational study nested in the national influenza surveillance system	South Africa	2007–2013	ILI outpatients of all ages; national influenza surveillance system; cell culture (before 2009) and RT-PCR (in and after 2009)	Selection criteria not described	140 (43 H1N1pdm09 38 H3N2, and 42 B); Sanger; South Africa (local)	NA	Mutations: drug sensitivity; antigenic analysis (NAI)	3, 4
Wadegu (2016) [105]	Observational study nested in a sentinel surveillance network	Kenya	2008–2011	ILI outpatients aged ≥ 2 months; sentinel site surveillance network; RT-PCR	Archived samples; selection criteria not described	92 (21 H3N2, 18 H1N1, and 53 H1N1pdm09; Sanger; Kenya (local)	HA (HA1), MP (M2)	Mutations; drug sensitivity; phylogenetics (Bayesian); phylodynamics; reassortment	3, 4

**H1N1** = seasonal H1N1 influenza, **H1N1pdm09** = pandemic (collected in 2009–2010) and seasonal H1N1pdm09 influenza (since 2011), and **H3N2** = seasonal H3N2 influenza, **ILI** = influenza-like illnesses, **SARI** = severe acute respiratory illnesses, **ARI** = acute respiratory infections, **LRTI** = lower respiratory tract infection, **HA** = hemagglutinin, **NA** = neuraminidase, PA, PB1, and PB2= polymerase subunits, **MP** = matrix protein (M1 and M2), **NS** = non-structural protein, **WG or WGs** = whole genome or whole genomes (consists of all the 8 gene segments in order of PB2, PB1, PA, HA, NP, NA, MP, and NS), **PGs** = partial genomes, phylogenetic reconstruction methods: **ML** = maximum likelihood, **NJ** = neighbor-joining, and **Bayesian**: MrBayes or BEAST, antigenic (serology) analysis tests: **HAI** = hemagglutination inhibition test, **NAI** = neuraminidase inhibitors, **IC_50_** = half maximal inhibitory concentration, **RT-PCR** = real-time reverse transcription polymerase chain reaction, **SH** = south hemisphere, **NH** =northern hemisphere, **WHO** = World Health Organization, **GISAID** = global initiative on sharing all influenza data, **NCBI** = National Center for Biotechnology Information, **GenBank** = nucleic acid sequence data bank, **GISRS** = global influenza surveillance and response system, **NIC or NICs** = National Influenza Centre(s), **IRD** = influenza resource database, **U.K.** = United Kingdom, **subs/site/year** = substitutions per site per year. Patient variables: **1** = sex, **2** = age, **3** = location, **4** = time, **5** = HIV status, **6** = other/co-infection, **7** = ethnicity, **8** = symptoms/severity, **9** = travel, **10** = pregnancy, **11** = participation in another study/trail, **12** = household, **13** = oseltamivir treatment. Severity was inferred as having pneumonia, admission to intensive care facilities or death.

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
