# Peer review of "Molecular Epidemiology and Evolutionary Dynamics of Human Influenza Type-A Viruses in Africa: A Systematic Review"

_microorganisms, 2022, doi:10.3390/microorganisms10050900_

Round 1

Reviewer 1 Report

Nabakooza et al present “Molecular epidemiology and evolutionary dynamics of human influenza type-A viruses in Africa: a systematic review”, a review delving into the ecology of H1N1 and H3N2 viruses in Africa through sequence data from published articles.  The study is highly descriptive, and at times difficult to follow due to the sheer volume of descriptions.  Nevertheless, the study is well executed and very well written for the most part, for this the authors should be commended.

General

The authors are assigning countries of origin to viruses, despite these being simply one isolate of a circulating lineage – please could the authors be more clear that for example “A/Johannesburg/141/2007” might be a South African strain, but it is not necessarily a South African virus, unless its clear origins circulating in Africa at previous dates are shown.

The authors should consider a graphical way to describe section 3.4, which would increase the number of readers that could digest this article.

The authors should consider adding subtype information to strain names (e.g. A/Michigan/35/2015 (H1N1), to increase the clarity when sections switch between H1N1 and H3N2.

Line by line:

Line 41, perhaps “potentially deadly” is more suitable

Line 42+, also Swine IAVs

Line 49, please refine, as proteins such as M2 are not really ‘internal’, this sentence could be more accurate

Line 51, this again is poorly phrased, many subtypes infect humans, but the authors are suggesting these subtypes are the only ones that have been able to circulate in/spread between humans, rather than from zoonotic spill over.

Line 58, drift away from vaccine mediated immunity

Line 138, “one African virus”?  Or specify

Line 215, please clarify what is meant by “H1N2 viruses became extinct shortly”

Line 233, “similar to the vaccine virus”?

Some typos concerning Côte d'Ivoire in the MS, this is the correct spelling, or I would suggest the authors use the English name “Ivory Coast”

Line 386, can the authors further describe what is meant by oscillating genetic diversity

Author Response

REVIEWER 1:

COMMENTS AND SUGGESTIONS FOR AUTHORS

Point 1: Nabakooza et al present “Molecular epidemiology and evolutionary dynamics of human influenza type-A viruses in Africa: a systematic review”, a review delving into the ecology of H1N1 and H3N2 viruses in Africa through sequence data from published articles.  The study is highly descriptive, and at times difficult to follow due to the sheer volume of descriptions.  Nevertheless, the study is well executed and very well written for the most part, for this the authors should be commended.

Response 1: Thank you for commending our efforts, study methodology, and manuscript presentation. Yes, we acknowledge that descriptive results can sometimes be hard to follow. Therefore, we shortened or paraphrased some sentences or paragraphs throughout the manuscript for clarity. Furthermore, we complemented most of the hard-to-follow sections like 3.4.4 and 3.5.3 (circulating clades among Africa H1N1pdm09 and H3N2 strains, respectively) with figures and tables as appropriate (Main text Figures 1-5 and Supplementary S8-S12).

GENERAL

Point 2: The authors are assigning countries of origin to viruses, despite these being simply one isolate of a circulating lineage – please could the authors be more clear that for example “A/Johannesburg/141/2007” might be a South African strain, but it is not necessarily a South African virus, unless its clear origins circulating in Africa at previous dates are shown.

Response 2: The terms “virus” or “viruses” were replaced with “strain” or “strains” throughout the manuscript, respectively when referring to viruses sampled in a specific geographical location (household or city or country) at a given time.

Point 3: The authors should consider a graphical way to describe section 3.4, which would increase the number of readers that could digest this article.

Response 3: Thank you for the intriguing suggestion. It was the most extensive part of the revision but worth it. Each subtype (H1N1, H1N1pdm09, and H3N2) has two major subsections 1) their Genomic diversity of H1N1 viruses and their relatedness to vaccines and 2) phylogenetic clustering and circulating clades.

For the virus-vaccine comparisons, the most relevant data extracted was on the a) % gene and protein similarity and b) amino acid substitutions with significance such as those that cause antigenic drift or alter virulence, transmissibility, pathogenicity, severity, and drug sensitivity. Unfortunately, several studies reported substitutions without associating them with any biological function. Therefore, we lacked sufficient relevant data to report per country over the sampled period. Furthermore, influenza vaccine formulations are updated according to strains circulating per season (year), and most geographical regions share the same formulation. Therefore, we described virus-vaccine changes according to year rather than geographical (per location). Describing temporal trends seems an appropriate way to appreciate viral evolution and the continuous antigenic drift to justify the need to update vaccine strain formulations per year (season).

Fortunately, several studies classified sampled virus strains into global genetic clades (based on the characteristic substitutions in their HA1 proteins, ECDC protocol). We revisited each study that did the phylogenetic analysis and clade classification on any number of Africa strains (as low as one). We recorded the clade to which any Africa strain(s) belonged, whether it was originally sampled/sequenced by that study or not. For example, given a Ugandan study sampled/sequenced 40 H1N1 strains and included one strain from Cameroon all belonging to clade X, we reported the circulation of clade X in both countries at that given time. This way, we obtained sufficient data and a better resolution of circulating clades even for under-sampled countries where only one strain was analysed. Sections on circulating clades of Africa virus strains (H1N1, H1N1pdm09, and H3N2) were supplemented with Figures (3-5) and Tables (S8, S9, and S11) as appropriate.

Point 4: The authors should consider adding subtype information to strain names (e.g. A/Michigan/35/2015 (H1N1), to increase the clarity when sections switch between H1N1 and H3N2.

Response 4: Subtypes (H1N1 or H3N2) were added for each strain name mentioned throughout the manuscript.

LINE BY LINE:

Point 5: Line 41, perhaps “potentially deadly” is more suitable

Response 5: Replaced “deadly” with “potentially deadly” on page 1 line 40.

Point 6: Line 42+, also Swine IAVs

Response 6: Term “swine” added in the statement on page 2 line 42.

Point 7: Line 49, please refine, as proteins such as M2 are not really ‘internal’, this sentence could be more accurate

Response 7: Statement was refined to include genes coding for the 2 surface, 1 transmembrane, and 5 internal proteins on page 2 line 48-51.

Point 8: Line 51, this again is poorly phrased, many subtypes infect humans, but the authors are suggesting these subtypes are the only ones that have been able to circulate in/spread between humans, rather than from zoonotic spill over.

Response 8: The word “only” was replaced with “frequently” to mean that H1-H3 and N1-N2 subtypes circulate among humans at higher incidence/prevalence than the rest of the subtypes. Please see page 2 line 51.

Point 9: Line 58, drift away from vaccine mediated immunity

Response 9: Phrase “vaccines and escape host immunity” was replaced with “vaccine-mediated immunity” on page 2 line 59.

Point 10: Line 138, “one African virus”?  Or specify

Response 10: Previously, we had excluded two global studies (Barr et al., 2010, 2014) because they had analysed at most one strain sequence sampled in only 1-3 African countries (one strain per country). Specifically, the first study by Barr et al. (2010) included two strains (A/Johannesburg/37/2008(H1N1)-clade 2B and A/Kenya/1428/2008(H3N2)-clade 2) in their global H1N1pdm09 virus clade classification. The second study by Barr et al. (2014) included 4 strains(A/Dakar/20/2012(H1N1), A/Tunisia/159/2013(H1N1)-clade 6, A/Kenya/3294/2011(H1N1)-clade 6, and A/Tanzania/2074/2012(H1N1)-clade 7) in their global H1N1pdm09 virus clade classification. Barr et al., 2014 also included 5 strains (A/Johannesburg/5223/2012(H3N2)-Clade 3A, A/Cairo/140/2012(H3N2)-clade 3B, A/Cairo/136/2012(H3N2)-Clade 3B, A/Mauritius/549/2012(H3N2)-Clade 3C, and A/Dakar/4/2012(H3N2)-Clade 3C) in their global H3N2 virus clade classification.

However, given the unstable nature of RNA viruses, any strain could carry potentially pandemic substitutions, substitutions that alter the virus functionally (antigenicity, virulence, pathogenicity, and transmissibility), and give rise to novel viral lineages. Therefore, detecting and tracking the source and spread of any virus strain is critical in molecular epidemiology. Such knowledge could help in the early mitigation of circulating strains, especially during epidemics and pandemics, and provide scientific evidence for vaccine strain selection.

Since our major objective was to merge all existing data to the best of our ability and update our knowledge on influenza in Africa, we decided to include both studies in our analysis, bringing the total of eligible analysed studies to 71.  All 71 eligible studies were analysed regardless of the number of strains sampled. We are confident that we have tracked and shown all relevant data from even the under-sampled countries (sampling bias). Therefore, this statement “We excluded two articles (Barr et al., 2010, 2014) that analysed only one Africa virus, remaining with 69 articles included in the final analysis.” was deleted.

Point 11: Line 215, please clarify what is meant by “H1N2 viruses became extinct shortly”

Response 11: The new reassortant H1N2 strains did not circulate or were never sampled again shortly after their introduction in any human population in Africa. The previous statement “H1N2 viruses became extinct shortly” had been paraphrased for clarity on page 5 line 219-221.

Point 12: Line 233, “similar to the vaccine virus”?

Response 12: Corrected to “Similar to the A/California/07/2009(H1N1) strain” on page 6 line 245.

Point 13: Some typos concerning Côte d'Ivoire in the MS, this is the correct spelling, or I would suggest the authors use the English name “Ivory Coast”

Response 13: Spelling variants (CoteD’ivoire, CoteD’iviore, Cote D’Ivoire, and Cote D’iviore) were replaced with “Ivory Coast” throughout the manuscript.

Point 14: Line 386, can the authors further describe what is meant by oscillating genetic diversity

Response 14: We replaced the term “oscillating patterns” with “fluctuated continuously”. A fluctuating pattern has the high and low points. Please see page 9 line 413-414.  

Reviewer 2 Report

The authors carried a lot of work out to search information, summarize and systematize the data concerning influenza viruses (H1N1pdm09 and H3N2) and flu vaccines effectivity in Africa. The review is well done and written.

Unfortunately, virus spread and phylogenetic relationships between viruses circulated in different African countries is sometimes difficult to understand without illustration. It would be better to summarize and represent data as one phylogenetic tree (or schematics) with designation of viral clades for each subtype or as a table with information Clade/Years/Countries/Molecular Features/Reference. This material may be represented in Supplementary. This is only advice, but the authors may have own opinion.

            Figure 3 is not mentioned in a text of the article. Tabled 1 should be relocated into the Supplementary.

I have not received Figure S1 and Video S1.

            Misprints and remarks

  1. Abbreviations and terms should be unified:

H1N1pdm09 or pH1N1pdm09 (Lines 331, 389, 401, 680, 742, 756, 761 etc.)

Côte d'Ivoire or Cote d’Ivoire (or Ivory Coast) (Lines 334, 349, 503 etc.)

Different writing decimal digits

Line 258 etc.   ‘… of 24·55-35·77%, 39.6-41.8%, and 32.4-42.1% against the 2014-2016 and 2017…’

Abbreviations SARY (Line 289), SH, NH (Line 179), SNH (Line 460), BF (Line 611) are decoded only in the footnote for Table 1. It is advisable to do this earlier.

  1. Other remarks for following Lines (L)

L288    The sign before ‘312 I’ may have been omitted.

L310-311, 317, 350    ‘old viruses’

            Old or older viruses were sometimes isolated in the same year or half-year, therefore the word ‘early’ may be more suitable instead of ‘old’ virus?

L329-330        “Clade 7 continued to circulate in … and Kenya up to 2012…”

It does not coincide with Figure 3. According to the Figure 3 the H1N1pdm09 viruses absented from Africa in 2012.

L390-393 Uncomfortable to read.

“…While their MP, H1, NS, N1, PB2, PB1, PA, and NP genes evolved at a rate of 9.88x10-3 (5.58-14.5), 5.58x10-3 (2.75-9.28), 5.22x10-3 (1.64-9.17), 4.07x10-3 (1.47-7.73), 4.01x10-3 (1.47-6.45), 3.89x10-3 (1.29-7.22), 1.86x10-3 (3.03-6.19), and 0.80x10-3 (0.4-2.04) [52,53].”

            May it be replaced with another sentence?   

While their individual genes evolved at different mean rates such as MP - 9.88 x10-3 (5.58-14.5 x10-3), H1 - 5.58 x10-3 (2.75-9.28 x10-3), NS - 5.22x10-3 (1.64-9.17 x10-3), N1 - 4.07x10-3 (1.47-7.73 x10-3), PB2 - 4.01x10-3 (1.47-6.45 x10-3), PB1 - 3.89x10-3 (1.29-7.22 x10-3), PA - 1.86x10-3 (3.03-6.19 x10-3), and NP - 0.80x10-3 (0.4-2.04 x10-3) subs/site/year…

Line 599, 743 –HPD=3.09x10-3–5.31x10-3

            5.58x10-3(2.75-9.28) and 4.07x10-3(1.47-7.73) and 4.17x10-3(3.09-5.31)

Unify writing with above Lines 390-393

432      Is it right year 2017? Maybe, it is 2007?

624      H9_Ch/TUN/848_2011[109]

            Strain name does not correspond to common nomenclature. According to GenBank (JQ952591), this is the strain A/chicken/Tunisia/848/2011 (H9N2).

725, 738 – Reference is as an author name instead of list number.

740 -741   ’… ranges 5.0 (4.8-5.2x10-3, 4.4-5.34x10-3, and 3.8-5.21x10-3, respectively…’

Should be “…ranges 5.0 x10-3 (4.8-5.2x10-3, 4.4-5.34x10-3, and 3.8-5.21x10-3, respectively)…”

Figure 3, Footnote.    “HK=Hong Kong”  What is it? Strain name?

            MAD = Madrid/SO8171/2010            Should be ‘A/Madrid/SO8171/2010’

Author Response

REVIEWER 2: 

COMMENTS AND SUGGESTIONS FOR AUTHORS

Point 1: The authors carried a lot of work out to search information, summarize and systematize the data concerning influenza viruses (H1N1pdm09 and H3N2) and flu vaccines effectivity in Africa. The review is well done and written.

Unfortunately, virus spread and phylogenetic relationships between viruses circulated in different African countries is sometimes difficult to understand without illustration. It would be better to summarize and represent data as one phylogenetic tree (or schematics) with designation of viral clades for each subtype or as a table with information Clade/Years/Countries/Molecular Features/Reference. This material may be represented in Supplementary. This is only advice, but the authors may have own opinion.

Response 1: Thank you for the intriguing suggestion. It was the most extensive part of the revision but worth it.

We reported on the phylogenetic clustering/patterns among Africa IAV strains, where they clustered according to time (temporal evolution structure) and location (spatial evolution structure) of sampling. For the H1N1 (page 5 line 229), H1N1pdm09 (page 7 line 323-325), and H3N2 subtype (page 11 line 545-546).

Since genetic clades are inferred based on unique characteristic substitutions in the HA1 protein of the viruses, they are more important than phylogenetic clusters. Understanding the circulating clades is informative about the virus's genetic and antigenic nature which can direct vaccine strain selection. Furthermore, phylogenetic clustering patterns can be predicted if viral clades are known (circulating strains belonging to the same clade will cluster close together, although they might diverge by time or location of sampling). Therefore, we prioritized visualizing the genetic clades rather than phylogenetic clustering/relationships.

Fortunately, several studies classified sampled virus strains into global genetic clades (based on the characteristic substitutions in their HA1 proteins, ECDC protocol). To achieve the highest resolution of clades that circulated in different countries in Africa, we revisited each study that did the phylogenetic analysis and clade classification on any number of Africa strains (as low as one). We recorded the clade to which any Africa strain(s) belonged, whether it was originally sequenced/sampled by that study or not. For example, given a Ugandan study sampled/sequenced 40 H1N1 strains and included one strain from Cameroon, all belonging to clade X, we reported the circulation of clade X in both countries at that given time. This way, we obtained sufficient data and a better resolution of circulating clades even for under-sampled countries (with only one strain analysed). Sections on circulating clades of Africa virus strains (H1N1, H1N1pdm09, and H3N2) were supplemented with Figures (3-5) and Tables (S8, S9, and S11) as appropriate.

Point 2: Figure 3 is not mentioned in a text of the article. Tabled 1 should be relocated into the Supplementary.

Response 2: Figures were updated and mentioned as appropriate throughout the manuscript. The previous "Figure 3" was updated and transferred to the supplementary material as "Figure S10".

According to the PRISMA guidelines for reporting systematic reviews, Table 1 which shows the characteristics of studies analysed should be included in the main result not the supplementary. Therefore, we opt to keep Table 1 as is in the main text.

Point 3: I have not received Figure S1 and Video S1.

Response 3: There was no figure or video named S1 in our previous version. We just forgot to delete the default statement from the template. However, in the current revised manuscript, we renamed all supplementary Tables and Figures (S1-12) and mentioned them in the main text, as appropriate.

MISPRINTS AND REMARKS

Point 4: Abbreviations and terms should be unified:

H1N1pdm09 or pH1N1pdm09 (Lines 331, 389, 401, 680, 742, 756, 761 etc.)

Response 4: When we use H1N1pdm09 subtype constitutes both the pandemic (pH1N1pdm09) and seasonal (sH1N1pmd09). These are differentiated by the time they were sampled 2009-2010 for the pH1N1pdm09, and 2011 onwards for sH1N1pmd09. These were clearly defined under section 3.4 on page 5 line 234-236.

Point 5: Côte d'Ivoire or Cote d’Ivoire (or Ivory Coast) (Lines 334, 349, 503 etc.)

Response 5: Spelling variants (CoteD’ivoire, CoteD’iviore, Cote D’Ivoire, and Cote D’iviore) were replaced with “Ivory Coast” throughout the manuscript.

Point 6: Different writing decimal digits

Line 258 etc.   ‘… of 24·55-35·77%, 39.6-41.8%, and 32.4-42.1% against the 2014-2016 and 2017…’

Response 6: Decimal digits were unified on page 6 line 273.

Point 7: Abbreviations SARY (Line 289), SH, NH (Line 179), SNH (Line 460), BF (Line 611) are decoded only in the footnote for Table 1. It is advisable to do this earlier.

Response 7: Severe Acute Respiratory Illnesses (SARI) were defined on first mention on page 7 line 306. Northern hemisphere (NH) and 1988-1997 Southern Hemisphere (SH) also were defined on first mention on page 4 line 180-181.  Bayes Factor (BF) was defined on first mention on page 8 line 402-403.

OTHER REMARKS FOR FOLLOWING LINES (L)

Point 8: L288    The sign before ‘312 I’ may have been omitted.

Response 8: “312I’ this was not omitted, usually this signifies an insertion where a given virus (being compared to a reference sequence) had an amino acid Isoleucine (I) inserted at position 312 absent in the reference protein sequence (often the vaccine virus sequence).

Point 9: L310-311, 317, 350   ‘old viruses’

Old or older viruses were sometimes isolated in the same year or half-year, therefore the word ‘early’ may be more suitable instead of ‘old’ virus?

Response 9: “Old viruses” or “older viruses” was replaced with “earlier” on page 7 line 328, page 7 line 344, page 11 line 553, page 12 line 563.

Point 10: L329-330  “Clade 7 continued to circulate in … and Kenya up to 2012…”

It does not coincide with Figure 3. According to the Figure 3 the H1N1pdm09 viruses absented from Africa in 2012.

Response 10:  Figures (3-5, S10, S12), Tables (S8, S9, S11), and their corresponding descriptions for all sections on phylogenetic clustering and clades were harmonised as appropriate.  

Point 11: L390-393 Uncomfortable to read.

“…While their MP, H1, NS, N1, PB2, PB1, PA, and NP genes evolved at a rate of 9.88x10-3 (5.58-14.5), 5.58x10-3 (2.75-9.28), 5.22x10-3 (1.64-9.17), 4.07x10-3 (1.47-7.73), 4.01x10-3 (1.47-6.45), 3.89x10-3 (1.29-7.22), 1.86x10-3 (3.03-6.19), and 0.80x10-3 (0.4-2.04) [52,53].”

May it be replaced with another sentence?  

While their individual genes evolved at different mean rates such as MP - 9.88 x10-3 (5.58-14.5 x10-3), H1 - 5.58 x10-3 (2.75-9.28 x10-3), NS - 5.22x10-3 (1.64-9.17 x10-3), N1 - 4.07x10-3 (1.47-7.73 x10-3), PB2 - 4.01x10-3 (1.47-6.45 x10-3), PB1 - 3.89x10-3 (1.29-7.22 x10-3), PA - 1.86x10-3 (3.03-6.19 x10-3), and NP - 0.80x10-3 (0.4-2.04 x10-3) subs/site/year…

Response 11:  This was clarified on page 9 line 418-421.

Point 12: Line 599, 743 –HPD=3.09x10-3–5.31x10-3

 5.58x10-3(2.75-9.28) and 4.07x10-3(1.47-7.73) and 4.17x10-3(3.09-5.31)

Unify writing with above Lines 390-393

Response 12: Intervals and ranges were unified as appropriate, on page 13 line 656, line 817-818.

Point 13: 432      Is it right year 2017? Maybe, it is 2007?

Response 13: It’s the year 2007 (the 2007 SH season A/Wisconsin/67/2005 vaccine) as indicated on page 10 line 465.

Point 14: 624      H9_Ch/TUN/848_2011[109]

Strain name does not correspond to common nomenclature. According to GenBank (JQ952591), this is the strain A/chicken/Tunisia/848/2011 (H9N2).

Response 14: Strain name was corrected to A/chicken/Tunisia/848/2011(H9N2) on page 14 line 683.

Point 15: 725, 738 – Reference is as an author name instead of list number.

Response 15: References updated and unified throughout the manuscript.

Point 16: 740 -741   ’… ranges 5.0 (4.8-5.2x10-3, 4.4-5.34x10-3, and 3.8-5.21x10-3, respectively…’

Should be “…ranges 5.0 x10-3 (4.8-5.2x10-3, 4.4-5.34x10-3, and 3.8-5.21x10-3, respectively)…”

Response 16: Corrected on lines 815-816.

Point 17: Figure 3, Footnote.   “HK=Hong Kong” What is it? Strain name?

MAD = Madrid/SO8171/2010            Should be ‘A/Madrid/SO8171/2010’

Response 17:  Strain names were updated in current Figure S10. A/Madrid/SO8171/2010(H1N1) indicated in main text page 8 line 381 and Figure 4

Reviewer 3 Report

Grace Nabakooza and co-workers present a thorough analysis of human influenza A viruses (IAV) in Africa isolated and sequenced between 1993 and 2018. Their metaanalysis observes the PRISMA guidelines for systematic reviews and was registered on PROSPERO. All selected publications were quality-checked according the Newcastle-Ottawa Assessment Scale and the STROME-ID checklist.

For each of the three viruses seasonal H1N1, pandemic H1N1pdm09 and seasonal H3N2, the authors describe the molecular epidemiology, i.e., the accumulation of amino acid exchanges especially in the major antigenic sites in comparison to the respective vaccine strains, the antiviral drug sensitivity, potential N-glycosylation sites and phylogenetic clustering. Finally, H3N2-H1N1pdm09 reassortants and zoonotic infection were analysed. The results presented here are of interest for health authorities and molecular epidemiologists.

The study is well-performed and only minor corrections are necessary. These are:

1. line 5: delete "and"

2. line 167: This sentence lacks a verb.

3. line 233: Did you mean "... similar to the vaccine strain..."?

4. line 455/456: Did you mean non-agglutination of chicken red blood cells? Please specify.

5. line 621: Phylogenetic analysis of 102 human...

6. Authors do not refer to Figure 3 in the main text.

Author Response

REVIEWER 3:

COMMENTS AND SUGGESTIONS FOR AUTHORS

Grace Nabakooza and co-workers present a thorough analysis of human influenza A viruses (IAV) in Africa isolated and sequenced between 1993 and 2018. Their metaanalysis observes the PRISMA guidelines for systematic reviews and was registered on PROSPERO. All selected publications were quality-checked according the Newcastle-Ottawa Assessment Scale and the STROME-ID checklist.

For each of the three viruses seasonal H1N1, pandemic H1N1pdm09 and seasonal H3N2, the authors describe the molecular epidemiology, i.e., the accumulation of amino acid exchanges especially in the major antigenic sites in comparison to the respective vaccine strains, the antiviral drug sensitivity, potential N-glycosylation sites and phylogenetic clustering. Finally, H3N2-H1N1pdm09 reassortants and zoonotic infection were analysed. The results presented here are of interest for health authorities and molecular epidemiologists.

The study is well-performed and only minor corrections are necessary. These are:

Response: We appreciate such a great comment!

Pont 1: line 5: delete "and"

Response 1: Deleted please see page 1, line 4-5.

Pont 2: line 167: This sentence lacks a verb.

Response 2: sentence corrected on page 4 line 167.

Pont 3: line 233: Did you mean "... similar to the vaccine strain..."?

Response 3: Corrected to “similar to the A/California/07/2009(H1N1) vaccine strain” on page 5 line 242, 245

Pont 4: line 455/456: Did you mean non-agglutination of chicken red blood cells? Please specify.

Response 4: These were avian red blood cells as indicated on Page 10 line 491.

Pont 5: line 621: Phylogenetic analysis of 102 human...

Response 5: Paraphrased to “Phylogenetic analysis 102 virus sequences sampled in Tunisia from humans (H1, H3, N1, and N2)” on page 13 line 679-683.

Pont 6: Authors do not refer to Figure 3 in the main text.

Response 6: Figures were updated and mentioned as appropriate throughout the manuscript. The previous "Figure 3" was updated and transferred to the supplementary material as "Figure S10".
